# Facilitating two-electron oxygen reduction with pyrrolic nitrogen sites for electrochemical hydrogen peroxide production

Wei Peng[1], Jiaxin Liu[1], Xiaoqing Liu[1], Liqun Wang ®[2] ✉, Lichang Yin ®[3] ✉, Haotian Tan[1], Feng Hou ®[1] ✉ & Ji Liang ®[1] ✉

Electrocatalytic hydrogen peroxide ($H_2O_2$) production via the two-electron oxygen reduction reaction is a promising alternative to the energy-intensive and high-pollution anthraquinone oxidation process. However, developing advanced electrocatalysts with high $H_2O_2$ yield, selectivity, and durability is still challenging, because of the limited quantity and easy passivation of active sites on typical metal-containing catalysts, especially for the state-of-the-art single-atom ones. To address this, we report a graphene/mesoporous carbon composite for high-rate and high-efficiency $2e^-$ oxygen reduction catalysis. The coordination of pyrrolic-N sites -modulates the adsorption configuration of the *OOH species to provide a kinetically favorable pathway for $H_2O_2$ production. Consequently, the $H_2O_2$ yield approaches $30 \, \text{mol} \, g^{-1} \, h^{-1}$ with a Faradaic efficiency of 80% and excellent durability, yielding a high $H_2O_2$ concentration of $7.2 \, g \, L^{-1}$. This strategy of manipulating the adsorption configuration of reactants with multiple non-metal active sites provides a strategy to design efficient and durable metal-free electrocatalyst for $2e^-$ oxygen reduction.

Hydrogen peroxide ($H_2O_2$) is an extremely important chemical that has been widely used in a serial of industry and healthcare fields, such as disinfection, bleach, and water treatment[1–3]. In the post COVID-19 era, especially, the demand for $H_2O_2$ as a disinfectant will remain at a high level for a long time[4]. By far, the industrial production of $H_2O_2$ mainly follows the anthraquinone oxidation process (AOP), which suffers from high energy consumption and serious pollution[5,6]. Besides, the as-produced $H_2O_2$ is usually in a high-concentration form for reducing transportation costs, which will inevitably lead to additional safety concerns. Therefore, the development of a miniaturized, distributed, and energy-efficient process for $H_2O_2$ production is important and urgent for the sustainable development of $H_2O_2$-related industries. Considering this, the electrocatalytic two-electron oxygen reduction reaction ($2e^-$-ORR) has been regarded a promising alternative to the

traditional AOP due to its energy-efficient and environment-friendly features[7–9]. During the $2e^-$-ORR process for $H_2O_2$ production, the electrocatalysts play the key role, which fundamentally determines the selectivity, yield, and stability of the whole process. Consequently, the exploration of advanced electrocatalysts with these desirable features has become a critical issue in this field.

Benefiting from high catalytic activity and atomic utilization, single-site catalysts have been widely investigated for $2e^-$-ORR electrocatalysis[7,10]. For such catalysts, by adjusting the atomic configuration of the active sites, their surface electronic structures can be finely manipulated to effectively adsorb the *OOH species, which is the only intermediate in the $2e^-$-ORR process, thus efficiently accelerating the $2e^-$-ORR process. Generally, the single-site catalysts developed for $2e^-$-ORR electrocatalysis include metal (i.e., carbon materials

[1]Key Laboratory of Advanced Ceramics and Machining Technology of Ministry of Education School of Materials Science and Engineering, Tianjin University, Tianjin 300072, China. [2]Applied Physics Department, College of Physics and Materials Science, Tianjin Normal University, Tianjin 300387, China. [3]Shenyang National Laboratory for Materials Science, Institute of Metal Research, Chinese Academy of Sciences, Shenyang, Liaoning 110016, China. ✉ e-mail: wlxywlq@mail.tjnu.edu.cn; lcyin@imr.ac.cn; houf@tju.edu.cn; liangji@tju.edu.cn

decorated transition-metal atoms)[11–13] or non-metal ones (i.e., carbon materials doped with N, O, F, and S heteroelements)[14–16]. Compared with the single-site catalysts for electrocatalysis, those with multiple active centers (i.e., the multiple-site catalysts) have the potential for achieving superior catalytic performance because multiple-site catalysts can not only increase the number of active centers, but also more versatilely modulate the electronic structure of the material surface and effectively optimize the adsorption of various reactants and intermediates[17]. Due to these unique advantages, they have been utilized in a number of electrocatalysis processes, including four-electron oxygen reduction reaction (4e⁻-ORR), nitrogen reduction reaction (NRR), and carbon dioxide reduction reaction (CO₂RR)[14,18,19]. As for the 2e⁻-ORR, research mainly focused on the construction of active centers with closed coordinated multiple metal atoms, including homonuclear and heteronuclear metal atoms.

Despite these merits, the intrinsic issues of these metal-based multi-site catalysts may pose a significant impedance to the further enhancement of their catalytic capability. Firstly, similar to the metal-based single-site catalysts, the multiple-site catalysts are also susceptible to being passivated during electrocatalytic reactions, especially by the various oxygen-containing species[20]. In addition, the density of multiple-metal active centers on the catalyst surface can hardly achieve a sufficiently high level due to the easy aggregation of metal atoms. Consequently, although these metal-based multiple-site catalysts commonly show high activity for 2e⁻-ORR that leads to a very low overpotential and high selectivity, however, they are often unable to achieve satisfactory H₂O₂ yield rate and durability for 2e⁻-ORR electrocatalysis.

In this case, constructing multiple-site catalysts that are based on non-metal active centers may be a feasible solution to the above-mentioned issue[21,22]. On the one hand, it is much easier to achieve a high doping level for the multiple-non-metal active centers compared with the metal sites, without aggregation even under high temperatures[23]. On the other hand, the multiple-non-metal active centers are less likely to be passivated during the catalytic reaction or be etched off from the substrate, even under acidic environments, thus showing much superior durability than their metal-based counterparts. Due to these merits, the non-metal multiple-site catalysts (e.g., B-N and S-N dual doped carbons) have been adopted in a variety of electrocatalytic processes, such as 4e⁻-ORR, hydrogen evolution reaction (HER), and CO₂RR with excellent performance even comparable to the noble-metal catalysts[24–26], which demonstrates their potential feasibility for 2e⁻-ORR. Among the non-metal active sites, the pyrrolic N species have been proven to have a high 2e⁻-ORR selectivity because of its suitable adsorption strength with the *OOH species, which is just slightly lower than the ideal value (4.23 eV). Compared with other N doping configurations with 4e⁻-ORR properties (e.g., pyridinic N or graphitic N), which often possess excessively stronger adsorption for *OOH, the pyrrolic N configuration with relatively weaker adsorption should be a more suitable candidate for constructing the proposed multiple-site catalyst. This is because the cooperation of multiple doping sites often intensifies the adsorption of the *OOH species and thus may result in an ideal strength in the case of multiple pyrrolic N sites[19].

Based on these considerations, we herein present the synergy of multiple pyrrolic N sites decorated on the graphene/mesoporous carbon composite for high-performance 2e⁻-ORR. This unique synergy efficiently optimizes the adsorption configuration of the (intermediate) reactants and significantly reduces the kinetic barrier for 2e⁻-ORR, according to the explicit solution model and slow-growth approach investigations. Benefiting from this, a high H₂O₂ yield up to ca. 30 mol g⁻¹ h⁻¹ is achieved, yielding a high H₂O₂ concentration of 7.2 g L⁻¹ over 24 h, together with excellent selectivity and durability. Moreover, this strategy also has the potential to be extended to other non-metal multiple-site catalysts, offering a universal route for designing advanced catalysts for 2e⁻-ORR and other potential processes.

## Results

### Synthesis and characterizations of materials

The fabrication of the material is schematically illustrated in Fig. 1a. Specifically, F127, as the soft template for forming mesopores, and resol, as the carbon precursor, were firstly dissolved in ethanol. Melamine was subsequently added to the solution. $C_3N_4$ produced during high-temperature treatment of melamine as the nitrogen precursor and a second template to form graphene over the resol-derived mesoporous carbon[27–29]. Afterward, the mixture was dried under ambient conditions, followed by thermal curing. The obtained product was then annealed at 900 °C under N₂ protection to obtain the final product, denoted as P-NMG-X (X = 0, 5, 10, and 15, representing the ratio of melamine and resol). Details of the materials fabrication can be found in the Methods.

The morphology of the P-NMG-X samples was firstly characterized by scanning electron microscopy (SEM). As shown in Figure S1, the P-NMG-0 sample, prepared in the absence of melamine, exhibits a bulky and dense structure with a large particle size of up to tens of microns. With the increase of melamine dosage, P-NMG-10 and P-NMG-15 gradually show a loose structure, with twisted and wrinkled graphene nanosheets uniformly distributed among the porous carbon blocks (Figs. 1b and S2–5). The demarcation of mesoporous carbon and graphene can be clearly observed by the blue dashed line in Fig. 1b. This demonstrates that the introduction of melamine in the fabrication is also capable of changing the morphology of the mesoporous carbon and guiding the in-situ formation of graphene, in addition to acting as a precursor for nitrogen doping. Then the typical P-NMG-10 sample was investigated by transmission electron microscopy (TEM), which clearly reveals the abundant and disordered mesopores on the carbon blocks with a pore size of 10–20 nm (Fig. 1c, d). Besides, it can also be found that the graphene nanosheets are rooted in the mesoporous carbon blocks, both of which are basically amorphous without distinct graphitic lattice structures. The amorphous feature of the materials was also confirmed by XRD, which shows only one broad peak near ca. 23° (Figure S6)[30].

The porosity of the materials was studied by the nitrogen adsorption technique. The nitrogen adsorption-desorption isotherms of P-NMG-10 show a distinct hysteresis loop at medium to high-pressure regions (P/P₀ = 0.4–0.8, Fig. 1e), which can be attributed to mesopores and macropores. The pore diameter calculated by Barrett-Joyner-Halenda (BJH) method is ca. 20–50 nm, consistent with the TEM observations. In addition, a sharp rise exists in the isotherms in the low-pressure region (Fig. 1e), which corresponds to a large number of micropores in the materials (<2 nm). This is also in agreement with the pore size distribution results. The large number of microporous structures is beneficial for enhancing the specific surface area and the content of defects in the materials, which further affects the configurations of heteroatom doping and the catalytic properties of the materials. Besides, Fig. 1e shows that P-NMG-10 has a specific surface area of 529 m² g⁻¹, higher than P-NMG-5 (243.0 m² g⁻¹, Figure S7) and P-NMG-15 (526.3 m² g⁻¹, Figure S8). It is worth noting that the adsorption and desorption curves appear to be non-closed, which we believe may be caused by the flexible pore structure or N₂ chemisorption. On the one hand, this porous structure brings a large specific surface area, which is beneficial for the exposure of active sites during electrocatalysis. On the other hand, the unique pore structure of this material, with both large mesopores and macropores, can also promote the rapid diffusion of O₂ inside the catalyst as well as the desorption of H₂O₂ product, thus simultaneously accelerating the 2e⁻-ORR and preventing the excessive reduction of produced H₂O₂.

More structural information of the materials was then obtained by Raman spectroscopy, in which two typical bands can be found at 1300

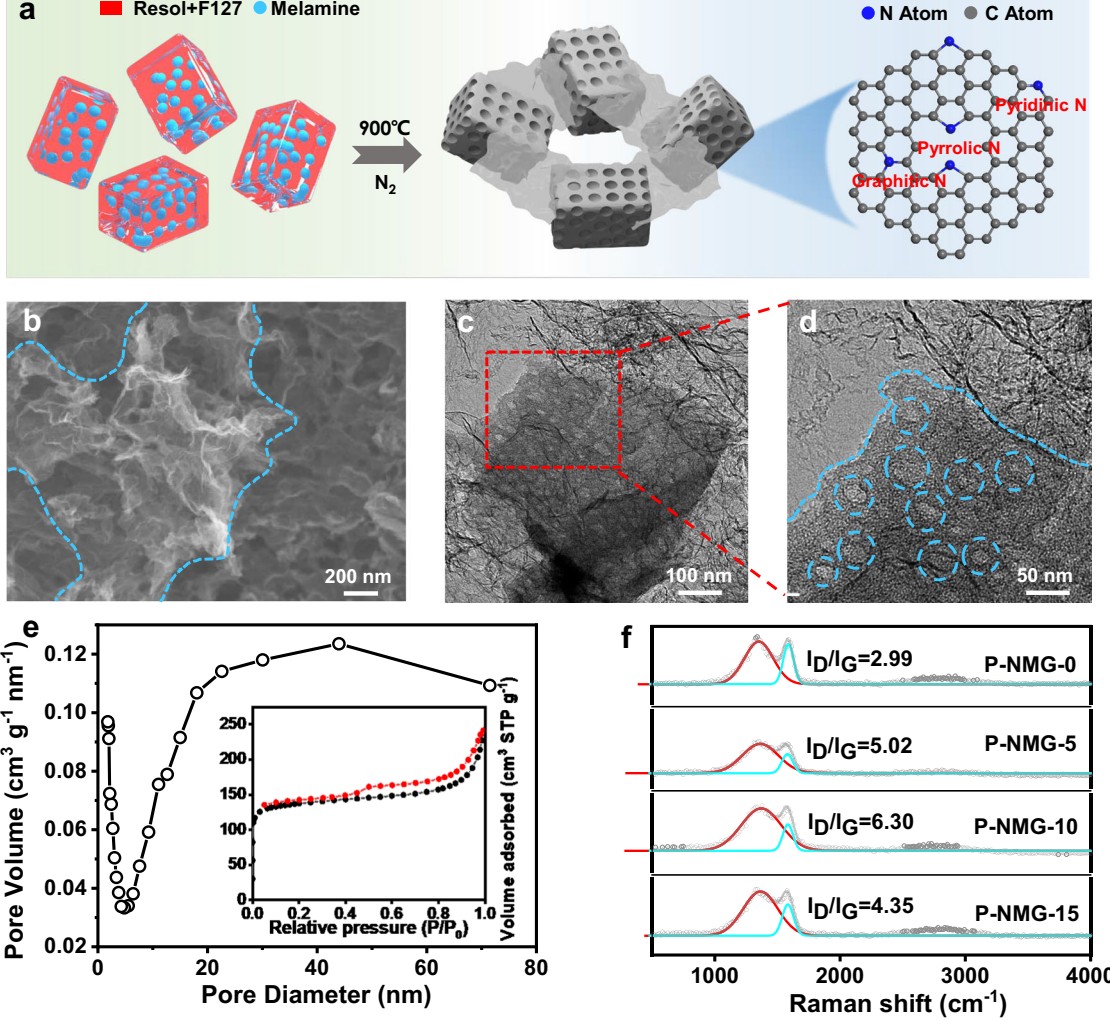

**Fig. 1 | Formation and microstructural characterizations of the materials.**
**a** Schematic illustration of the synthesis of P-NMG-X. **b** SEM image of P-NMG-10.
**c** Low-resolution and **d** high-resolution TEM images of P-NMG-10. **e** Nitrogen
adsorption-desorption isotherm, and the corresponding pore size distribution of P-NMG-10. **f** Raman spectra of the analogue samples.

and 1580 cm$^{-1}$ (Fig. 1f), representing the lattice/edge defects (i.e., the D band) and the in-plane stretching vibration of the sp$^2$ hybridized carbon atoms (i.e., the G band) in the carbon framework, respectively. The intensity ratios of the D and G bands suggest the defect content of the material. By fitting the Raman data to the respective bands[29], it can be found that P-NMG-10 has the highest $I_D/I_G$ ratio of 6.3, showing its highest defect content among the P-NMG-X samples. The largest quantity of defects in P-NMG-10 may be associated with its abundant pore structure introduced through the F127 soft template.

The chemical composition of the materials was firstly investigated using TEM-based energy dispersive X-ray spectroscopy (EDS). For the typical P-NMG-10 material, the EDS elemental mappings demonstrate the homogeneous distribution of C, N, and O elements over its surface (Fig. 2a), suggesting the uniform nitrogen doping of the porous carbon. Afterward, X-ray photoelectron spectroscopy (XPS) was carried out. Similar to the EDS results, C, N, and O elements can be found in the survey scan of P-NMG-5,10 and 15 (Figure S9). With the increase of melamine dosage in the fabrication, the N contents in the obtained materials also increased (0, 8.69, 9.38, and 9.47 at.% for P-NMG-0, 5, 10, and 15, Fig. 2b). It can be seen from Fig. 2b that the oxygen content in P-NMG-0 is much higher than that in P-NMG-5/10/15. The highest content for P-NMG-0 should be attributed to its bulky structure without large pores for the release of oxygen-containing species during high

temperature anneal. The relatively lower oxygen content for P-NMG-5/ 10/15 may be due to their abundant macropores as well as the melamine precursor that releases ammonia and other nitrogen-containing species as possible reduction agents to help remove oxygen from carbon. The fine chemical states of these elements were then studied by high-resolution XPS. As shown, the N 1s spectra of the materials can be deconvoluted into three peaks located at 398.3, 399.8, and 401.1 eV, which can be assigned to pyridinic-N, pyrrolic-N, and graphitic-N species, respectively (Fig. 2c–e)[31]. For these materials, the content of the pyrrolic N firstly increases and then decreases, with the increased melamine dosage in the precursor. Specifically, the relative ratio of pyrrolic N reached 50% for P-NMG-10, corresponding to an atomic ratio of 4.7 at.%, much higher than that of P-NMG-5 (23%, 2.0 at.%) and P-NMG-15 (26%, 2.5 at.%, Fig. 2f). Besides, the C 1s spectra show three peaks at ca. 284.5, 285.2, and 288.5 eV, which can be assigned to three carbon species, including C=C, C=N/C−N, and C−O, respectively (Figure S10)[32]. In addition, the high-resolution O 1s spectra are deconvoluted into three peaks, corresponding to C=O (531.0 eV), C−O −C/O=C−O (532.4 eV), and C−OH (533.7 eV)[33]; and their contents are on a similar level (Table S1 and Figure S11). By fitting the split peaks to the oxygen fine spectra, there is no strong correlation between the oxygen-containing functional groups and the catalytic properties, so we speculate that oxygen may not be involved in the catalytic reaction.

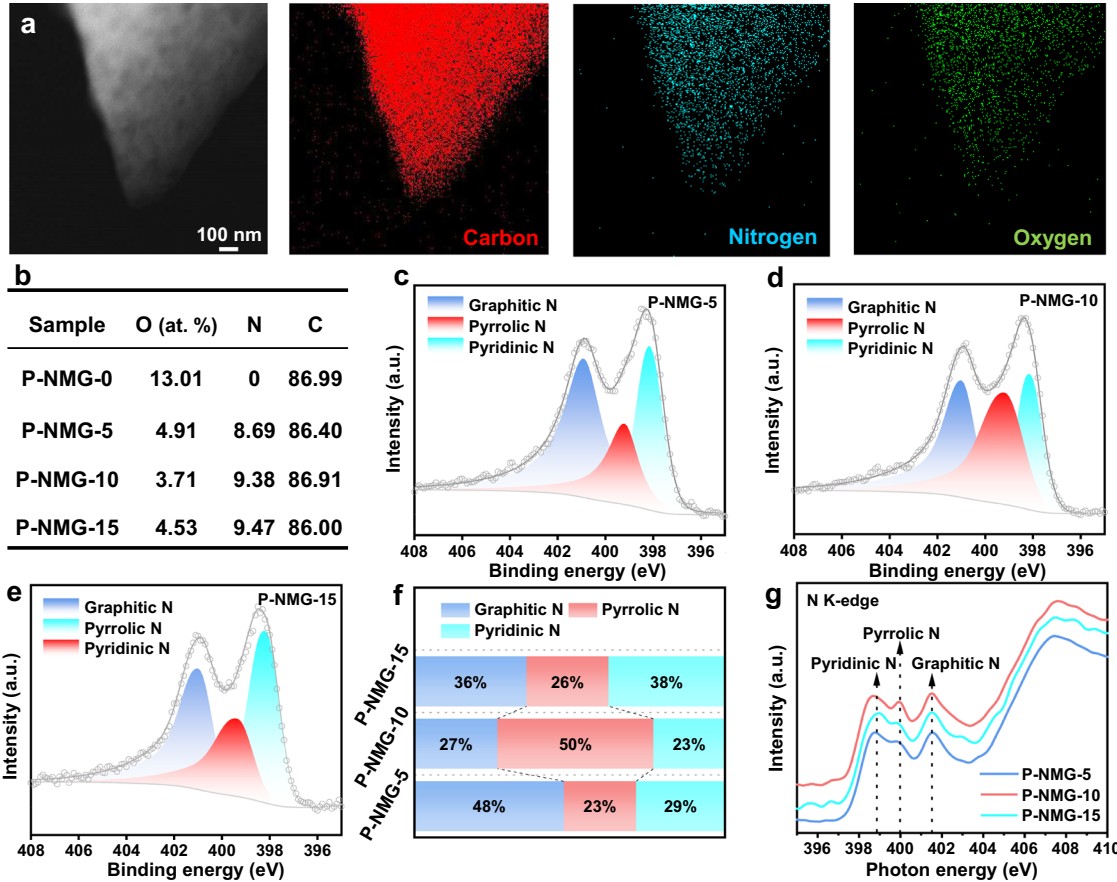

**Fig. 2 | Chemical structure characterization and XANES measurements of P-NMG-X. a** TEM image and the corresponding TEM-EDS elemental mapping of P-NMG-10. **b** Atomic percentages of P-NMG-X analogues. N *1s* XPS spectra for **c** P-NMG-5, **d** P-NMG-10, and **e** P-NMG-15. **f** The relative contents of graphitic nitrogen, pyrrolic nitrogen, and pyridine nitrogen species in the materials. **g** N K-edge XANES spectra of the materials.

The FTIR spectra of the samples show a broad band centered at 3430 cm$^{-1}$ corresponding to the N−H stretching vibrations from pyrrolic nitrogen structure (Figure S31). The peak at 1025−1200 cm$^{-1}$ is assigned to C−N bonds in the molecule. These two FTIR characteristic peaks of P-NMG-10 material is higher than that of P-NMG-5/15, which proves that P-NMG-10 has a higher content of pyrrolic nitrogen. The presence of different N species was also confirmed using N K-edge X-ray absorption near edge structure (XANES) spectra. By normalizing the data, it can be seen that the peak at 400.0 eV, which corresponds to the pyrrolic N species, is significantly more intensive for P-NMG-10 than the P-NMG-5 and P-NMG-15 (Fig. 2g)[34]. It again clearly proves that P-NMG-10 has the highest pyrrolic N content, in good alignment with the XPS results. In addition, ICP-MS characterization of P-NMG-10 was performed in order to exclude the possible interference of trace metals in the material for subsequent electrochemical tests. The results of Table S3 demonstrate that P-NMG-10 does not contain common transition metals and noble metals.

The highest defect content for P-NMG-10 should be the compound result of several factors. On the one hand, the material is composed of two types of carbon structures, including the porous carbon blocks derived from phenolic resin and the none pores graphene-like carbon nanosheet (Figure S5). As the amount of melamine increases, the thickness of the porous carbon blocks decreases and the amount of none pores graphene-like carbon nanosheet increases. On the second hand, the melamine particles of a few microns in size, which embeds in the phenolic resin precursor to form a composite precursor for the subsequent anneal, also acts a template to form large pores in the porous carbon blocks in the finally obtained

material. These macropores can not only affect the specific surface area and defect content of the material, but also promote the efficient removal of the F127 soft template during annealing. Consequently, P-NMG-0, forming large carbon blocks without any macropores, possess only a very low specific surface area as well as a small content of micropores. In contrast, the carbon blocks in P-NMG-15, with the most abundant macropores, should possess the highest amount of micropores in its carbon block sections. However, the large portion of none-porous graphene-like nanosheet may compromise its overall pore amount. Thus, P-NMG-10, with both a high content of porous carbon blocks and a high porosity of porous carbon blocks, possesses the highest overall amount of micropores, i.e., the most sufficient defect sites for the formation of pyrrolic nitrogen species. Pyrrolic nitrogen, with five-membered ring structure, exists only at the edge of the carbon framework or around the pores inside the carbon framework, i.e., the defect sites on carbon. Therefore, the content of pyrrolic nitrogen species tends to be proportional to the defect content of the carbon material. Subsequent chemical structure characterization results demonstrated the highest pyrrolic nitrogen content corresponding to the highest defect level of P-NMG-10.

## Electrocatalytic 2e$^-$-ORR characterizations

The electrocatalytic ORR performance of the materials was then evaluated in an 0.1 M KOH aqueous electrolyte. Figure 3a shows the cyclic voltammetry (CV) curves of P-NMG-10 obtained in the Ar- and O$_2$-saturated electrolytes, respectively. In the O$_2$-saturated electrolyte, the CV curve of P-NMG-10 shows a distinct ORR current at ca. 0.6 V vs. reversible hydrogen electrode (RHE), which is absent in the

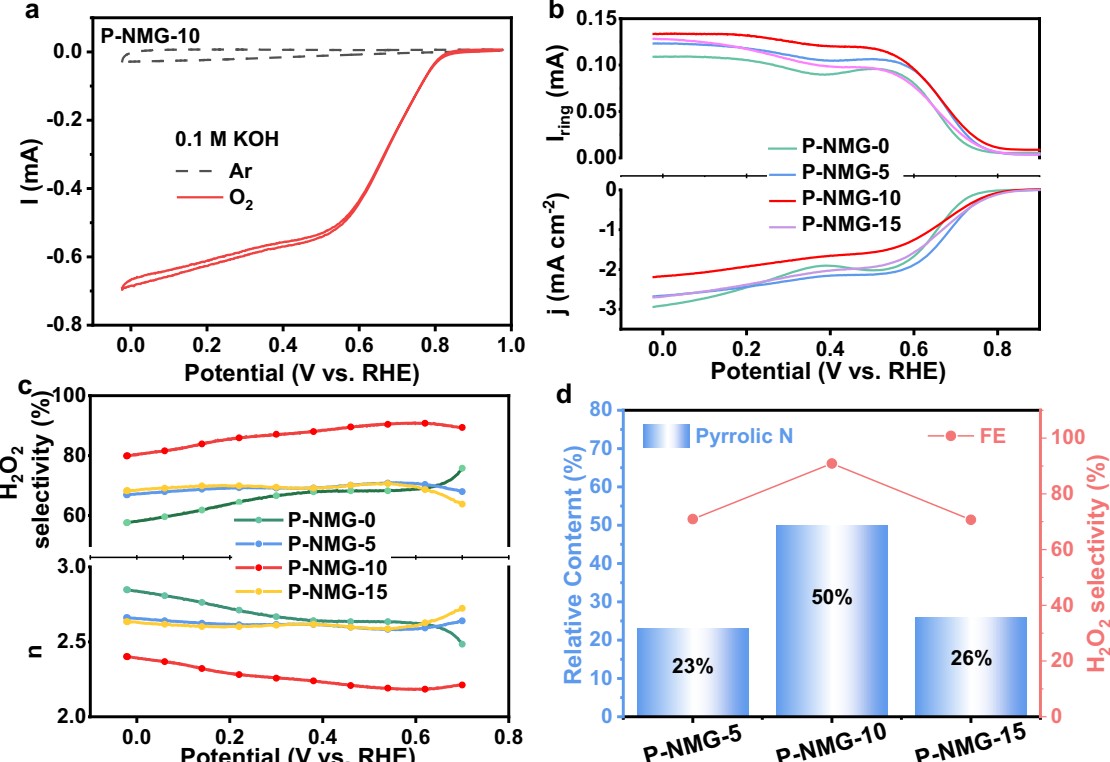

**Fig. 3 | Electrochemical 2e⁻-ORR catalytic performance of P-NMG-X materials.**
**a** CV curves of P-NMG-10 in $O_2$ and Ar-saturated 0.1 M KOH solution at a scan rate of
10 mV s⁻¹. **b** LSV curves of different samples, tested at 1600 rpm and a rate of
5.0 mV s⁻¹, showing the current (densities) on the disk ($j_{disk}$) and ring ($I_{ring}$) elec-
trodes. **c** $H_2O_2$ selectivity and electron number of the materials. **d** Relationship
between the pyrrolic N content and the selectivity of $H_2O_2$.

Ar-saturated electrolyte, indicating the capability of the material for
catalyzing ORR[33]. Subsequently, linear sweep voltammetry (LSV) was
conducted on a rotating ring-disc electrode to further assess the 2e⁻-
ORR performance of the materials. As shown in the corresponding LSV
curves of the materials (Fig. 3b), P-NMG-10 has the largest ring current
and the smallest disk current, indicating that it is more inclined to the
2e⁻-ORR pathway. The $H_2O_2$ selectivity and electron transfer number of
the different samples were plotted versus the testing potential
(Fig. 3c)[35]. As shown, the $H_2O_2$ selectivity of P-NMG-10 was ca. 91%,
corresponding to an n value of ca. 2.2, which is constantly the highest
among the samples in the whole potential range. Meanwhile, for
P-NMG-10, a disk current density of 1 mA cm⁻² was achieved at 0.64 V
vs. RHE, which is close to the equilibrium potential for 2e⁻-ORR
(i.e., 0.7 V vs. RHE), representing a facile ORR kinetics with negligible
overpotential for $O_2$-to-$H_2O_2$ conversion[36–38].

By comparing the relation between the pyrrolic N content and the
selectivity of 2e⁻-ORR for the P-NMG-X analogues (Fig. 3d), it is clear
that the increase in the pyrrolic N content of the material can improve
its selectivity for 2e⁻-ORR. To further confirm this, we also tested the
materials' activity for $H_2O_2$ reduction reaction ($H_2O_2$RR) in the Ar-
saturated electrolyte containing 1 mM $H_2O_2$. Again, P-NMG-10 has the
lowest activity for $H_2O_2$RR (Figure S14), which is beneficial for the
retention of high $H_2O_2$ production by preventing its further reduction.
The above electrochemical tests clearly demonstrate that P-NMG-10
has excellent 2e⁻-ORR selectivity, which is positively correlated with its
high pyrrolic N contents.

The $H_2O_2$ yield of P-NMG-X was then evaluated in an H-type cell.
In order to maximize the electrocatalytic reaction triple interface,
the reaction was performed using P-NMG-X-coated carbon paper as
the working electrode, a Pt plate as the counter electrode, and a
Nafion film to separate the cathode/anode chambers[39]. The $H_2O_2$
yield of the catalyst was obtained by subtracting the $H_2O_2$ yield of

the carbon paper for the same time from the total yield after the
test. The accumulated $H_2O_2$ concentration in the electrolyte was
quantified by the potassium oxalate colorimetric method and the
standard curve is shown in Figure S15. The experiments of the blank
control groups excluded the possibility of the oxygen feeding gas
and the material itself causing the colorimetric of the electrolyte
(Figure S16).

The yield and FE of different samples at different potentials show
that all samples had the highest FE value at 0 V vs. RHE. Therefore, the
yields of different samples at 0 V vs. RHE were selected for compar-
ison. As shown in Figs. 4a and S19, the $H_2O_2$ yield and FE of P-NMG-10
are higher than that of P-NMG-0, P-NMG-5, and P-NMG-15. We then
investigated the $H_2O_2$ yield for P-NMG-10 in a wider potential region
down to −0.8 V vs. RHE. In this case, the highest value is achieved at
−0.6 V vs. RHE, with an $H_2O_2$ yield rate up to ca. 30 mol g⁻¹ h⁻¹ (611.4
μmol h⁻¹ with a FE of 80%), significantly superior to the comparative
analogues over the whole testing potential range (Fig. 4b). Remark-
ably, a high FE of 81% values can still be achieved in this case, clearly
showing the selectivity of this material for 2e⁻-ORR.

To better understand the reason for the high $H_2O_2$ yield under this
very negative potential, LSV was carried out in a wider potential range
from −0.6 to 0.8 V vs. RHE under both $O_2$ and Ar atmospheres (Fig. 4c).
As shown, P-NMG-10 exhibits a very high overpotential for the HER in
the Ar atmosphere. As a result, in the $O_2$-saturated electrolyte, stable
current plateaus could be achieved on both the ring and disk for up to
−0.6 V vs. RHE, clearly suggesting its high selectivity against not only
4e⁻-ORR but also HER. In addition, electrochemically active surface
area (EASA) of the materials was also assessed, which shows P-NMG-10
has the highest EASA among the analogues (Figure S20). Conse-
quently, this high $H_2O_2$ yield and excellent FE of P-NMG-10 can be
reasonably attributed to its large EASA for 2e⁻-ORR as well as the strong
inhibition against the undesirable 4e⁻-ORR and HER.

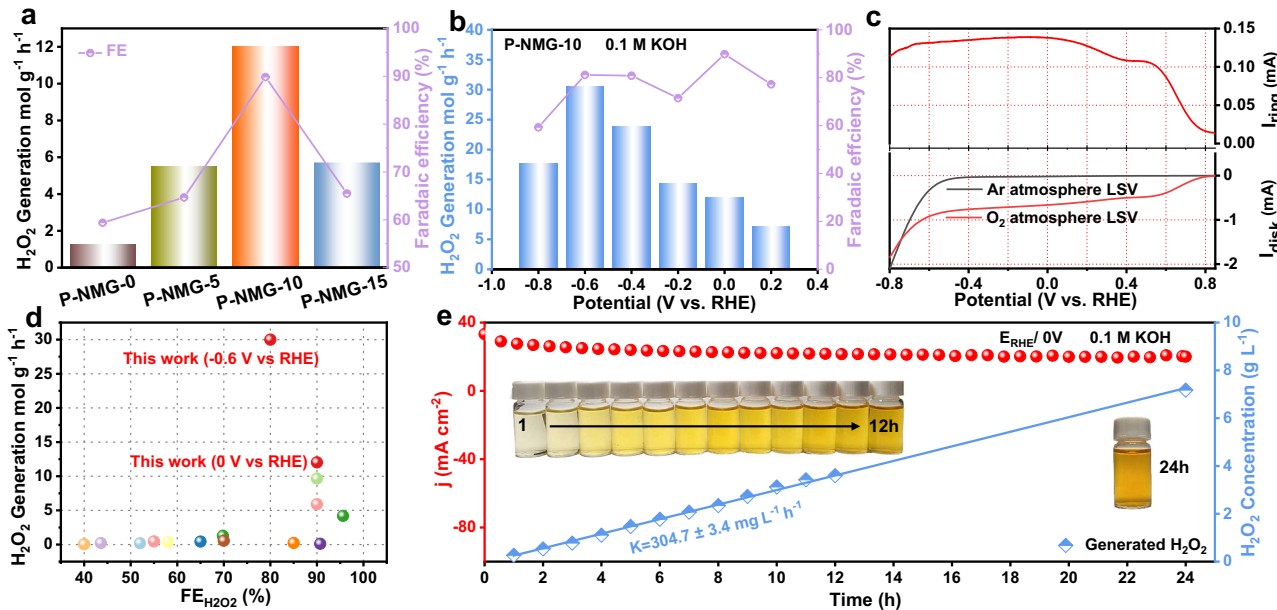

**Fig. 4 | 2e⁻-ORR catalytic performance of P-NMG-X materials in the two-compartment cell. a** $H_2O_2$ yield of the materials tested at 0 V vs. RHE. **b** The $H_2O_2$ yield of P-NMG-10 at different potentials. **c** The long-range LSV curves of P-NMG-10 in $O_2$ and Ar-saturated electrolytes. **d** Comparison of the $H_2O_2$ yield and FE of P-NMG-10 with some of the reported samples. **e** The chronoamperometry testing curves for 24 h and linear fitted lines for the change in $H_2O_2$ concentration.

To better illustrate the performance of this material, it is then compared with the recently reported 2e⁻-ORR catalysts in terms of $H_2O_2$ yield and FE (Fig. 4d and Table S2). The $H_2O_2$ yield of P-NMG-10 is one of the highest among the materials reported so far. In addition, the stability of P-NMG-10 for electrocatalytic 2e⁻-ORR was also evaluated and the accumulative concentration of $H_2O_2$ in the electrolyte was tested every 1 h (Figure S22). A continuous 24 h non-stop i-t test was conducted. For P-NMG-10, the ORR current is fairly stable during the whole 24 h testing period. The electrolytes after different reaction periods were subjected to chronoamperometry reactions, and the significant increase of hydrogen peroxide concentration with time can be seen in the yellowish vials in inset of Fig. 4e. An average $H_2O_2$ concentration increasing rate of 304.7 mg L⁻¹ h⁻¹ and accumulative $H_2O_2$ concentration up to 7.2 g L⁻¹ were achieved in the H-cell, which is one of the highest reported values by far (Fig. 4e). In addition, 24 h non-stop i-t tests in the flow-cell achieved a final cumulative $H_2O_2$ concentration of 9 g L⁻¹ (Figure S34), with an average $H_2O_2$ concentration increasing rate of 383.3 mg L⁻¹ h⁻¹, slightly higher than that of the H-cell. These test results together demonstrate the excellent performance of P-NMG-10 in fabricating $H_2O_2$ by 2e⁻-ORR. Different concentrations of $H_2O_2$ were left in 0.1 M KOH and no significant decrease in $H_2O_2$ concentration was observed even when left for up to 24 h (Figure S29). In addition, several rounds of 30 min repeatability tests were conducted, the material exhibited stable yields and FE in all 12 rounds of testing (Figure S23). Both of the above experiments demonstrated the good stability of P-NMG-10. Apart from conventional 0.1 M KOH electrolyte, the $H_2O_2$ yield rate of the material was also assessed under neutral conditions. To evaluate its feasibility for practical medical disinfection applications, we used 0.9 wt% aqueous NaCl solution as the electrolyte, rather than the typical phosphate-buffered saline (PBS). In this case, P-NMG-10 again exhibited a high $H_2O_2$ yield of 11 mol g⁻¹ h⁻¹ at −0.2 V vs. RHE and excellent stability (Figures S24–25), clearly demonstrating its viability for various targeted applications.

## Theoretical calculations

The content of carbon nitride varies with the amount of melamine, and the content of carbon nitride can significantly affect the structure of carbon substrate morphology and the process of nitrogen doping. On this basis, the soft templating method generates sufficient pores and structural defects in the carbon framework. Different levels of holes and graphene are present in the structure of the composite carbon substrate, and defects at the edges are the only environment where pyrrolic N exists[40]. Considering the results of the carbon substrate structure and chemical characterization, when the ratio of C atoms to pyrrolic N atoms reaches 20:1, adjacent multiple pyrrolic N structures should be highly likely to present on the carbon substrate, forming the expected clusters containing multiple pyrrolic N sites for 2e⁻-ORR. A detailed discussion can be found below Figure S12 in the supplementary information. In this case, it may also exhibit a similar synergistic effect toward ORR, as widely observed in the other metal-based multiple-site systems[14]. Currently, the presence and content of pyrrolic nitrogen can be measured and quantified by available chemical structure characterization techniques (e.g., XPS, FTIR, and XANES)[31,34]. However, due to the limitation in the current material characterization techniques, it is extremely difficult or even impossible for us to directly observe the multiple pyrrolic nitrogen sites on the materials. It is expected that advances in aberration-corrected transmission electron microscopy or scanning tunnel microscopy technologies may help directly probe the multiple pyrrolic nitrogen sites at the atomic scale in the future and greatly facilitate the study of heteroatom-doped materials with low relative atomic masses[41,42]. To better understand this positive relation of material's performance with its high pyrrolic N contents, density functional theory (DFT) calculations were performed to obtain the adsorption free energy of key intermediates in ORR by an implicit solvation method. Meanwhile, the ab initio molecular dynamics (AIMD) simulations were also conducted in the alkaline environment to describe the reaction kinetics.

Graphene with dual-pyrrolic nitrogen doping configuration (Dual-PyrN-Gr) was constructed as a simplified model to simulate multiple pyrrolic N sites that may be presented in P-NMG-10. We have simulated three possible structures, all of which have very low overpotentials by theoretical calculations. In order to facilitate the subsequent calculations, we chose the first structure with the lowest

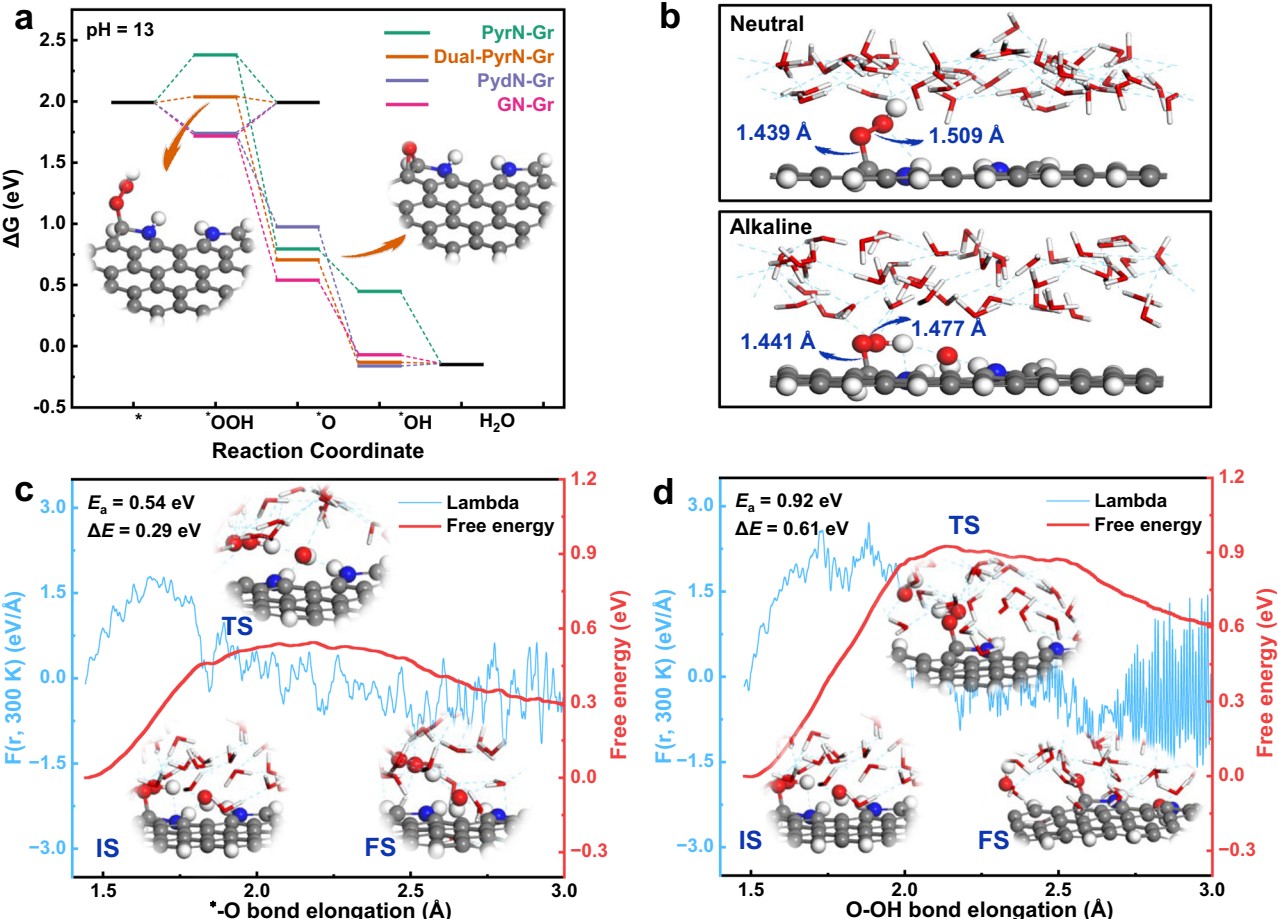

**Fig. 5 | Theoretical results on the 2e⁻-ORR activity on different configurations.** **a** Free energy profile of 2e⁻ and 4e⁻-ORR pathways at the equilibrium potentials of 0.7 V vs. RHE on different N-doped graphene models. **b** Schematic diagram of *OOH adsorption on Dual-PyrN-Gr in neutral and alkaline solutions under equilibrium potentials. Free energy changes as a function of CV along. **c** C−O bond and **d** O−O bond cleavage for the 2e⁻ and 4e⁻-ORR pathways, respectively. The initial state, transition state, and final state structures are denoted as IS, TS, and FS, respectively. Blue, gray, red, and white spheres represent the N, C, O, and H atoms, respectively.

overpotential as the best model (Figure S26). For comparison purposes, sole-pyrrolic N-doped graphene (PyrN-Gr), sole-pyridinic N-doped graphene (PydN-Gr), and sole-graphitic N-doped graphene (GN-Gr) were also considered (Figure S27). The free energy profiles for both 2e⁻ and 4e⁻-ORR pathways are given in Fig. 5a. As shown, Dual-PyrN-Gr presents a near zero overpotential towards the formation of $H_2O_2$ ($\Delta G_{*OOH} = 0.05$ eV) for the 2e⁻-ORR pathway. In contrast, all the PyrN-Gr, PydN-Gr, and GN-Gr counterparts require much higher overpotentials (0.39, 0.25, and 0.27 V, respectively) for the 2e⁻-ORR process. This result is well consistent with our experimental observation. In order to exclude the possible influence of oxygen functional groups in the calculation, we introduced three different oxygen functional groups for the calculation based on the Dual-PyrN-Gr doping configuration (Figure S33). The thermodynamic calculation results demonstrate that the 2e⁻-ORR overpotential of the dual-pyrrolic nitrogen doping configuration site is not affected by the oxygen functional groups (Table S4).

To better understand its superior 2e⁻-ORR performance, the electronic structure of Dual-PyrN-Gr was analyzed by taking three N-doped graphene nanoribbons with different widths as examples. Figure S28 shows the spin-polarized partial density of states (PDOS) for the $2p_z$ electrons of the neighboring C atom of pyrrolic N in both Dual-PyrN-Gr and PyrN-Gr with or without the *OOH adsorption. As we can see, obvious peaks appear around the Fermi level for the three considered Dual-PyrN-Grs without the *OOH adsorption, which disappears upon the adsorption of *OOH. However, no obvious PDOS peaks can

be observed for the three considered PyrN-Grs with or without the *OOH adsorption. This clearly indicates that the higher PDOS value around the Fermi level for the Dual-PyrN-Gr than that of PyrN-Gr should be the main reason for the superior 2e⁻-ORR performance of the Dual-PyrN-Gr. In fact, as widely demonstrated, materials with higher PDOS values around the Fermi level commonly possess a higher chemical reactivity or catalytic activity[43–46].

It should be noted, however, that the DFT results based on the implicit solvation model indicate a more energetically favorable 4e⁻-ORR pathway than the 2e⁻-ORR pathway for all considered N-doped graphene models (Fig. 5a), which seems contrary to our experiment observations (Fig. 3c). As we know, the implicit solvation model cannot accurately simulate the realistic solid-liquid environment, thus failing to describe the subtle structure change of the intermediates adsorbed on the catalyst surface[45]. On the contrary, the explicit solvation model can have a significant impact on the local environment at the solid-liquid interface, thus can well describe the structure change of the ORR intermediates absorbed on the catalyst surface[47,48], which may significantly change the kinetic behavior of ORR. Therefore, we re-optimized the geometry structure of the *OOH absorbed on Dual-PyrN-Gr in the neutral and alkaline aqueous solution by using the explicit solvation model (details about this method can be found in the Supplementary Information). As we can see in Fig. 5b, within the frame of the explicit solvation model, after we introduce the OH group, the C−O bond becomes longer and the O−O bond becomes shorter. This

change in bond length may lead to an easier breaking of the C−O bond, implying a more kinetically favorable 2e⁻-ORR pathway.

To confirm this, we further calculated the reaction barriers of the Dual-PyrN-Gr material for the decisive step using an explicit solution of molecular dynamics method under the alkaline environment[49]. For the 2e⁻-ORR pathway, the reaction barrier was estimated to be 0.54 eV by the slow-growth approach (Fig. 5c)[50]. In contrast, if the reaction proceeded via the 4e⁻-ORR pathway, the reaction barrier formed by *O was found to be 0.92 eV, which is significantly higher than that of the 2e⁻-ORR pathway (Fig. 5d). Since the 4e⁻-ORR pathway requires higher energy to dissociate the strong O−O bond, the Dual-PyrN-Gr thus exhibits a lower selectivity towards the 4e⁻-ORR pathway and is more favorable for the formation of $H_2O_2$ in kinetics. Therefore, this result based on the explicit solution model combined with the slow-growth approach, while taking into account a real alkaline environment, comprehensively and clearly shows that the 2e⁻-ORR pathway is more favorable than the 4e⁻-ORR pathway for our materials, which is in good agreement with the experimental results.

## Discussion

In summary, the graphene/mesoporous carbon composites doped by multiple pyrrolic N sites have been successfully synthesized, in which the synergy of pyrrolic N sites can effectively modulate the adsorption configuration of the *OOH species; meanwhile, the reaction barrier of 2e⁻-ORR can be greatly reduced, which provide a kinetically favorable pathway for $H_2O_2$ production. Consequently, an exceptionally high $H_2O_2$ yield up to 30 mol g⁻¹ h⁻¹, a satisfactory FE of 80%, and excellent durability for a continuous operation during 24 h can be obtained, respectively. Besides, a high $H_2O_2$ concentration of 7.2 g L⁻¹ can also be achieved. We hope that this strategy of multiple non-metal active sites to regulate the adsorption configuration of the reactant can shed light on the development of highly efficient, durable, and low-cost metal-free electrocatalysts for 2e⁻-ORR toward application-oriented $H_2O_2$ production.

## Methods

### Synthesis of resol

Typically, 20 g of phenol (Aladdin Biochemical Technology, Analytical Reagent) was heated until melted at 50 °C in a sealed bottle. Then 4.26 g of 20 wt% NaOH (Meryer Biochemical Technology, Analytical Reagent) aqueous solution was added by drop with a continuous stir. Afterward, 34.5 g of 37 wt% formaldehyde (Meryer Biochemical Technology, Analytical Reagent) solution was added by drop, and the heating temperature was raised to 70 °C. The solution was heated and stirred for another 1 h, and then cooled to room temperature. After that, the pH value of the solution was adjusted with 2 M HCl (Meryer Biochemical Technology, Analytical Reagent) aqueous solution until 7. Then, water was removed by rotary evaporation and the product was diluted into a 20 wt% EtOH (Meryer Biochemical Technology, Analytical Reagent) solution. During the dilution, the separated NaCl (Meryer Biochemical Technology, Analytical Reagent) was then filtered, resulting in a pale yellow solution.

### Synthesis of P-NMG-X

0.6 g F127 (Macklin Biochemical Technology, average Mn-12000) was dissolved in 10 mL EtOH with 0.375 g 0.2 M HCl at 35 °C with continuous stirring for 30 min. Then 3 g of the above-prepared 20 wt% resol solution was added and kept stirred for another 20 min. Subsequently, 0, 1:5, 1:10, and 1:15 ratios (0, 3, 6, and 9 g) of melamine (Aladdin Biochemical Technology, Analytical Reagent) were added to the solution. Subsequently, the mixture was dried overnight at room temperature. The obtained solid was heated at 100 °C in the air for another 24 h. Then, P-NMG-X was prepared following the heating program: room temperature–60 °C (ramp rate: 1 °C min⁻¹, kept for 2 h)–600 °C (ramp rate: 1 °C min⁻¹)–900 °C (ramp rate: 5 °C min⁻¹, kept

for 2 h)–room temperature. The obtained materials were ground in a mortar and washed with deionized water and EtOH, and then dried.

### Characterizations

Micromeritics ASAP 2460 Automatic specific surface and porosity analyzer BET were utilized for obtaining the specific surface area, pore sizes and nitrogen adsorption-desorption curve of the material. The K-edge X-ray absorption spectra of N were measured on the soft X-ray spectroscopy beamline at the Hefei Synchrotron. Bruker D8 advanced X-ray diffraction (XRD) with Cu Ka radiation was used for analyzing the crystalline structures of the as-prepared materials. Hitachi S-4800 scanning electron microscopy (SEM) and Hitachi JEM-2100f transmission electron microscopy (TEM) were utilized to carry out morphology characterizations. Kratos Axis Ultra DLD X-ray photoelectron spectroscopy (XPS) with Al Ka X-rays was utilized for the determination of the surface chemical states of materials. Horiba Labram HR Evolution Raman was used for analyzing the defect content.

### Electrochemical measurements

Electrochemical measurements were conducted in a cell with the three-electrode configuration using a CHI 760E electrochemical workstation. Graphite rod electrode and Ag/AgCl (filled with saturated KCl solution) electrode were used as the counter electrode and the reference electrode, respectively. And an RRDE (disk area: 0.2475 cm²) with a Pt ring (ring area: 0.1866 cm²) as the working electrode. To prepare the catalyst ink, 4.0 mg of the obtained catalyst powders were dispersed in 970 μL of the mixture of isopropanol and water (v/v = 1:3.85) and 30 μL of 5 wt% Nafion solution. 5.0 μL of the catalyst ink was drop-casted on the disk electrode for the RRDE measurement. CV curves were tested under saturated oxygen and argon, respectively, with a sweep rate of 5 mV s⁻¹ and an interval of 0 to −1.8 V (vs. Ag/AgCl). The electrolyte was first purged with $O_2$ and an LSV curve was recorded in $N_2$-saturated 0.10 m KOH at the rotating speed of 1600 rpm. The ring currents were recorded by fixing the ring potential at 1.2 V vs. RHE to detect the $H_2O_2$ produced on the disk electrode. The collection efficiency (N) was immersed (Figure S10) in Ar-saturated 0.1 M Phosphate Buffer Saline with 5 mM of potassium ferricyanide ($K_3Fe(CN)_6$). The selectivity and electron transfer number are calculated as follows:

$$H_2O_2\% = 200 \times \frac{I_R}{I_D + I_R/N} \tag{1}$$

$$n = 4 \times \frac{I_D/N}{I_D + I_R/N} \tag{2}$$

The electrochemically active surface area was measured by the double-layer capacitance method. CV scans were conducted at the potential window from −0.05 to 0.05 V vs. Ag/AgCl reference electrode with scan rates of 5, 10, 15, 20, and 25 mV s⁻¹. By plotting the $(J_a–J_c)/2$ at 0 V vs. Ag/AgCl against the scan rate ($J_a$ is the anodic current density and $J_c$ is the cathodic current density), the slope value was calculated to be the double-layer capacitance ($C_{dl}$). Since the electrolyte concentrations in the electrochemical tests were all higher than 0.1 M, we did not perform iR-compensation.

### $H_2O_2$ yield test

The $H_2O_2$ yield test was carried out in a gastight H-type cell that is separated by a proton exchange membrane (Nafion 117). The membrane was pretreated before use. Platinum plate electrode and Ag/AgCl (filled with saturated KCl solution) electrode were used as the counter electrode and the reference electrode, respectively. 5.0 μL of the catalyst ink (4 mg mL⁻¹) was drop-casted onto the surface of the working electrode (0.5 × 1 cm² of hydrophobic carbon paper). The

electrolyte was 30 mL of 0.1 M KOH and the i-t test was performed for 30 min at different potentials (0.2, 0, −0.2, −0.4, −0.6, and −0.8 V vs. RHE), using an aeration bar to continuously drum oxygen onto the surface of the carbon paper. At the end of the test, the electrolyte was collected. The $H_2O_2$ concentration in the electrolyte was tested by the potassium titanium oxalate colorimetric method. The FE was calculated as follows.

$$FE = 100\% \times \frac{H_2O_2(mol) \times 2 \times 96485}{total\ charge\ passed\ (C)} \qquad (3)$$

Color development of potassium titanium oxalate: Prepare 0.05 M potassium titanium oxalate, 3 M sulfuric acid, and different concentrations of hydrogen peroxide solution. Take 3 mL of hydrogen peroxide solution, 0.5 mL of potassium titanium oxalate solution, 0.5 mL of sulfuric acid solution and 1 mL of deionized water to form a mixed solution. The absorbance was tested using UV-vis. The absorbance and the concentration of hydrogen peroxide were then fitted to obtain the standard curve. The $H_2O_2$ yield of the catalyst was obtained by subtracting the $H_2O_2$ yield of the carbon paper for the same time from the total yield after the test.

### DFT methods

In this work, all DFT calculations and AIMD simulations are conducted using the VASP package[51]. Structure relaxation and molecular dynamics were carried out based on spin-polarized DFT using the general gradient approximation (GGA) and Perdew-Burke-Ernzerhof functional[52–54]. The kinetic energy cutoff of plane wave was set to be 500 eV and the convergence criterion for the residual forces and total energies were set to be 0.02 eV/Å and $10^{-6}$ eV, respectively. The dispersion correction was described by DFT-D3 method[55,56]. The Gamma-centered Monkhorst-Pack k-points mesh with a density of 0.04 $Å^{-1}$ for structural relaxation and 0.02 $Å^{-1}$ for electronic structure calculations was used to sample the Brillioun zone[57].

Compared with armchair graphene nanoribbon (AC-GNR), zigzag graphene nanoribbon (ZZ-GNR) is metallic and has higher stability[58–61]. Taking these into consideration, we choose zigzag graphene nanoribbons as the structure model to construct the N-doped graphene with different nitrogen dopants (Figure S21).

We applied the VASPsol package to evaluate solvation effects predicted by an implicit solvent and the properties of water are described with a dielectric constant of $\epsilon_b = 80$[62,63].

The free energy for the reaction intermediate is defined as:

$$G = E_{DFT} + E_{ZPE} - TS \qquad (4)$$

Where the $E_{DFT}$ is the electronic energy calculated by DFT using the implicit solvation model, $E_{ZPE}$ denotes the zero point energy estimated within the harmonic approximation, and TS is the entropy at 298.15 K.

To evaluate the kinetic barriers, we used a constrained AIMD method, namely, the "slow-growth" approach to obtain the free-energy profile at 300 K[64–66]. A cutoff energy of 400 eV, a k-mesh grid of $1 \times 1 \times 1$, a time step of 0.5 fs, and a canonical ensemble (NVT) were adopted in the AIMD simulations. The whole system contains 45 $H_2O$ molecules (two layers of water molecules on the surface of N-doped graphene) and one $OH^-$ anion in the bulk water to simulate the alkaline environment. In this case, the $OH^-$ concentration is high enough to be close to the experimental pH conditions. As for the slow-growth method, the value of the reaction coordinate (namely ξ) linearly changes from the characteristic value at the initial state (IS) to that at the final state (FS) with a velocity of transformation ξ. The free energy

required to fulfill the transformation from IS to FS can be computed as:

$$w_{IS \to FS} = \int_{\xi(IS)}^{\xi(FS)} \left(\frac{\partial F}{\partial \xi}\right) \times \xi dt \qquad (5)$$

Theoretically, for an infinitesimally small ξ, the work $w_{IS \to FS}$ corresponds to the free-energy difference between the final state and initial state.

In this work, a collective variable (CV) called ξ was assigned to describe the geometric property along the reaction path, and the Lambda is the gradient ($\frac{\partial F}{\partial \xi}$) of the free energy F along the ξ. To describe the C−O (O−O) breaking process, the bond length of the C−O (O−O) bond was used as CV, and an increment of 0.0005 Å per AIMD step was applied to drive the chemical reaction.

## Data availability

The data that support the findings of this study are available from the corresponding authors upon reasonable request. Source data are provided with this paper.

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

## Acknowledgements

This work was supported by the National Natural Science Foundation of China (No. 22179093) and Tianjin University International Education Program for Outstanding Doctoral Thesis (No. C2-2021-004).

## Author contributions

W.P. performed most of the experiments, analyzed the data, and wrote the initial draft of the manuscript. J.X.L. performed the theoretical calculations. X.L. and H.T. performed some material characterization. F.H., L.W., and L.Y. took part in analysis and writing. J.L. supervised the study and co-wrote the paper. All the authors discussed the results and reviewed the manuscript.

## Competing interests

The authors declare no competing interests.
