## [Peer Review File · Nature Communications]

Facilitating Two-electron Oxygen Reduction with Pyrrolic Nitrogen Sites for Electrochemical Hydrogen Peroxide ProductionReviewers' comments:

Reviewer #1 (Remarks to the Author):

This manuscript reports N-doped porous carbon materials for the electrochemical production of H₂O₂. The reported catalytic activity seems to be highly significant, but the manuscript is not convincing in terms of the reason for the selective catalysis. Please see the following comments.

1. The electrochemical production of H₂O₂ was proposed by Yamanaka in 1990. The following paper should be mentioned.

[10.1016/0013-4686(90)87004-L]

As to the porous N-doped carbon, the following papers also should be mentioned.

[10.1021/ja300038p]

2. The equilibrium potential for the 2-e reduction is 0.7 V, but LSV voltammograms in Fig 3 shows even higher onset potentials. The authors should explain the reason for that. Figure 3c shows very high selectivity to H₂O₂ at 0.7 V, but this is theoretically very difficult because the over potential for the 2-e pathway is 0 V, whereas that for the 4-e pathway is over 0.5 V.

3. Ordinary N-doped carbon is known not to be selective to the 2-e pathway in alkaline media.

[10.1016/j.coelec.2020.08.015]

On the contrary, the authors report a very high selectivity to the 2-e pathway in alkaline media. This point should be more carefully clarified. The authors propose multiple pyrrolic nitrogen sites but the existence of such sites are not supported by experimental data. As to the synthetic method, the authors use melamine for the N source, but there is no particular reason for the formation of such N species.

4. The authors use the DFT study to explain the high catalytic activity, but this part is not convincing. The proposed active sites have not been supported by experimental data, and this method does not consider the effect of pH.

Reviewer #2 (Remarks to the Author):

In this work, Peng et al. reports the synthesis of a new carbon based nanomaterial for the electrocatalytic O₂ reduction. Apparently the work claims high H₂O₂ formation rates, but I have doubts about the reliability of the experiment. A current density of 40 mA cm⁻² is extremely high, very strange for a purely carbon based catalyst, and it is very odd that such high currents are not reflected in the RRDE tests, where rather typical current density are observed, even though the catalyst is the same. Why such a big discrepancy? Moreover, the electrocatalytic experiments are incomplete: first of all, there is some confusion in the chronoamperometric experiment for the 24 h: in the Experimental part it

is reported that the i-t curves are recorded for 30 min experiments, this is a very short time; so, are the CA experiments effectively run for 24 h continuously or it is 30 min experiment repeated many times? This is important to know because it has been reported that the selectivity may change after the initial short time. Then, a control experiment is missing to check the contribution of the supporting WE. Finally, the study focuses on one material only, where melamine was the N-doped carbon precursor, but the authors have not checked other substrates to see if this high activity arises uniquely from the melamine precursor. For example, in a 2018 study, Iglesias et al. (Chem 2018, 4, 106) used carbon nanohorns as the template and several N-containing carbon precursor to see the differences in activity. Other issues arise from the characterization of the material. Melamine is known to evolve to graphitic carbon nitride after high T treatment, and in fact the 398 and 401 eV peaks are also typical of carbon nitride, the authors speculate on the presence of pyridinic and graphitic N, this could be in effect some carbon nitride. The distribution of the three signals is also rather typical, I cannot see how the activity is so much controlled here by the structure, especially because the FE is not any higher (actually even smaller) than many other studies. Xanes as well assign pyrrole, pyridine and graphitic, but standards are not reported: the XANES of the three molecules should be measured for undoubtedly assign the peaks. Solid state ^{15}N NMR could be of aid to check the N groups.

The conclusion that abundant mesopores are confirmed by TEM is a bit of a speculation, there is only one image with three spots that may be random structural features, it does not make a statistic to draw conclusions. Moreover, the pore distribution shows a curve that goes over a very large distribution of the pore dimension, there is not a peak. The N_2 isotherm also is recorded mistakenly, the loop does not close well, indicating problems in the measurements. The material is not fully characterized, a control experiment should check adventitious presence of metal impurities. I do not understand why different melamine loadings lead to similar N content.

In the table of comparison, it is not indicated to which materials the authors are comparing their activity, and some carbon based catalysts which are among the best catalysts are not even reported in the manuscript. Overall, this work leaves me with many doubts, and I think that more experimental work must be conducted to undeniably understand the factors that determine such high currents, surely at this stage publication in Nat comm cannot be recommended.

Reviewer #3 (Remarks to the Author):

In the manuscript the authors introduce N-doped carbon-based material, which was synthesized from resol and melamine at various ratios. The authors claim that the products of synthesis (achieved at 900 C and under N_2 atmosphere) differed in nitrogen content and chemical composition, which information was derived from HR XPS spectra. The material was characterized by a wide spectrum of experimental characterization techniques and described as graphene/mesoporous carbon composite. The Raman spectra indicate on very defective and functionalized carbon structure. The synthesized materials were used for catalysis of hydrogen peroxide production via 2eORR process. The authors claimed they observed record yield for hydrogen peroxide production reaching up to 30 mol/g h and sufficient Faradic efficiency (in alkaline media). They also provide theoretical calculations to support the claims of the study, that pyrrolic sites are responsible for this activity. Without any doubt the work is interesting addressing

hot-topic research focused on hydrogen peroxide production over carbon-based catalysts. On the other hand, there are many critical points, which have to be clarified before publication of this manuscript.

Characterization and elemental composition. The authors provide survey XPS spectra of the materials, but they do not provide elemental composition in the main text (just content on nitrogen content). They should comment particularly on oxygen content, which is quite high and pronounced also in the Raman spectra. They should also clarify, how the HR XPS spectra of N have been fitted (A peak position assignment is not clear. They refer to a work, where the peaks were assigned ad hoc. The authors must provide convincing peak assignment for N-doping in graphene. In addition, they should explain which procedure has been used for spectra deconvolution.). The authors have to estimate an error of the fitting. The role of oxygen and the variability in oxygen content (Table S1) should also be clarified. The authors have to provide FTIR spectra documenting suggested chemical composition of the materials. The authors must provide ICP-MS data and determine metal residues, which can significantly contribute to the material performance. They have to provide material characterization after its usage. (One additional point, there seems to appear a small peak corresponding to nitrogen in ..0 material (Fig. 2b). Could you rule out nitrogen presence in this material?)

Mechanistic insights: The authors do not provide any explanation or testable hypothesis concerning the evolution of various nitrogen forms. Why the material ..10 displays the highest content of pyrrolic nitrogen (double with respect of both “next-door” ..5 and ..15 materials)? This important finding is not clarified.

Performance: The authors state “The above electrochemical tests clearly demonstrate that P-NMG-10 has excellent $2e^-$ -ORR selectivity, associated with its high pyrrolic N contents.” The authors do not provide convincing data for this strong claim of causation. They just provide some data indicating on a correlation, which is not convincing taking into account the questions concerning material characterization stated above. The authors do not provide any data about catalyst reusability, which is absolutely critical to provide convincing evidence that the material behaves like a catalyst not being consumed during “catalysis”.

Calculations: The authors employ GGA functional (PBE) with an empirical dispersion term (D3). It is well known that GGA functionals significantly underestimate reaction barriers due to SIE. The author must provide more robust data using methodology/functional, which does not suffer from SIE. The authors have to estimate free energy error bars and comment on convergence of the provided AIMD simulations. Could you provide real water density used in the simulations?

Quality of presentation: The authors provide many features on figures, which are not explained or documented (e.g., Fig. 1b and 1d – there are highlighted areas with dashed blue line, which are not explained). Fig. 4d caption can provide link to Table S3 (the authors should provide complete overview of recent works dealing with hydrogen peroxide production over N-doped graphenes). What are the yellowish vials in inset of Fig. 4e?

Reviewer #4 (Remarks to the Author):

This paper reports the multiple pyrrolic N sites decorated on the graphene/mesoporous carbon composite for 2e-ORR. The authors have done many physical characterizations (TEM, XPS, XAS, etc) and DFT calculations to identify the active sites for 2e-ORR but the overall electrochemical activities are not very good compared to recent publications. For example, the authors claimed that the catalysts produce ultra-high yield & high FEs for H₂O₂ electrosynthesis. However, this is not a very notable result since the current density only observed ~ 30 mA cm⁻² with FE ~80 %. Many of publications ensure > 50 mA cm⁻² operation with > 90 % yields (e.g., 200 mA cm⁻² with FE ~ 90 %, *Angew. Chem. Int. Ed.*, 61, e202206544 (2022)). Also, the following publication reports similar concept related to the H₂O₂ production from pyrrolic-N sites: Li, L. et al. Tailoring Selectivity of Electrochemical Hydrogen Peroxide Generation by Tunable Pyrrolic-Nitrogen-Carbon. *Adv. Energy Mater.* 10, 2000789 (2020). Therefore, this reviewer suggests the authors to do more in-depth studies and submit it to a more specialized journal. Some more specific comments are listed below:

1. The FEs are not very high as recent publications. For example, 1. Lu, Z. et al. High-efficiency oxygen reduction to hydrogen peroxide catalysed by oxidized carbon materials. *Nat. Catal.* 1, 156–162 (2018). 2. Li, L. et al. Tailoring Selectivity of Electrochemical Hydrogen Peroxide Generation by Tunable Pyrrolic-Nitrogen-Carbon. *Adv. Energy Mater.* 10, 2000789 (2020). 3. Li, L. et al. Tailoring Selectivity of Electrochemical Hydrogen Peroxide Generation by Tunable Pyrrolic-Nitrogen-Carbon. *Adv. Energy Mater.* 10, 2000789 (2020). 4. Li, L. et al. Tailoring Selectivity of Electrochemical Hydrogen Peroxide Generation by Tunable Pyrrolic-Nitrogen-Carbon. *Adv. Energy Mater.* 10, 2000789 (2020). Since the current density & FEs are proportional to H₂O₂ yields, it's hard to say ~ 30 mA cm⁻² & 80 % of FEs ensure the ultrahigh yield.
2. If the authors would like to claim the ultrahigh yield, the 2e-ORR activities must be further studied in 0.5 M KOH or higher concentration to achieve higher current density
3. What about the electrochemical activities in acidic media, such as 0.5 M H₂SO₄? Because H₂O₂ is stable in acidic medium and it's more beneficial to do 2e-ORR in acidic medium, especially for carbon-based materials.
4. The H₂O₂ is not stable in a strong alkaline solution (0.1 M KOH) in nature thus it's hard to accumulate for a long time (even 2 hours). Thus, typically the electrolyte needs to be refreshed or requires additives (i.e., H₂O₂ stabiliser) to prevent H₂O₂ decomposition. I could not find any experimental details related to electrolyte refresh or stabilizer. How did authors address these problems?

Response to Reviewer 1#

Original comment: This manuscript reports a N-doped porous carbon materials for the electrochemical production of H₂O₂. The reported catalytic activity seems to be highly significant, but the manuscript is not convincing in terms of the reason for the selective catalysis. Please see the following comments.

Comment 1: The electrochemical production of H₂O₂ was proposed by Yamanaka in 1990. The following paper should be mentioned. [10.1016/0013-4686(90)87004-L]. As to the porous N-doped carbon, the following papers also should be mentioned. [10.1021/ja300038p]

Response: Firstly, we would like to thank the reviewer for his/her positive comments and valuable suggestions on this manuscript, which are very helpful in improving the quality of our work. We have carefully read these suggested publications, and we do agree with the reviewer that these literatures should be mentioned. We have added the suggested articles by the reviewer as references (No. 9 and 16) in order to further improve the quality of this work, as follows:

“Considering this, the electrocatalytic two-electron oxygen reduction reaction (2e⁻-ORR) has been regarded as a promising alternative to the traditional AOP due to its energy-efficient and environment-friendly features.⁷⁻⁹”

“Generally, the single-site catalysts developed for 2e⁻-ORR electrocatalysis include metal (*i.e.*, carbon materials decorated transition-metal atoms)¹¹⁻¹³ or non-metal ones (*i.e.*, carbon materials doped with N, O, F, and S heteroelements).¹⁴⁻¹⁶”

“9. Otsuka, *et al.* One step synthesis of hydrogen peroxide through fuel cell reaction. *Electrochim. Acta* **35**, 319-322 (1990).

16. Fellingner, T. P., *et al.* Mesoporous nitrogen-doped carbon for the electrocatalytic synthesis of hydrogen peroxide. *J. Am. Chem. Soc.* **134**, 4072-4075 (2012).”

Comment 2: The equilibrium potential for the 2e⁻ reduction is 0.7 V, but LSV voltammograms in Fig 3 shows even higher onset potentials. The authors should explain the reason for that. Figure 3c shows very high selectivity to H₂O₂ at 0.7 V, but this is theoretically very difficult because the over potential for the 2e⁻ pathway is 0 V, whereas that for the 4e⁻ pathway is over 0.5 V.

Response: Thanks for this valuable suggestion. The equilibrium potential for 4e⁻-ORR is 1.23 V *versus* RHE, and indeed at 0.7 V it may occur due to the large overpotential of 0.5 V. However, only

a very weak or even negligible $4e^-$ -ORR actually occurs at 0.7 V or lower potentials on our materials. This high selectivity for $2e^-$ -ORR against $4e^-$ -ORR is proven by not only RRDE results, but also the high H_2O_2 yield/FE as shown in the colorimetric tests. These results suggested that, at 0.7 V, although the overall reaction rate is low, the reaction still tends to be of a $2e^-$ -ORR pathway. We attribute this to the much lower kinetic barrier for $2e^-$ -ORR than that of the $4e^-$ -ORR counterpart at the multi-pyrrolic nitrogen sites on our material, which is discussed in the last part of the manuscript. In addition, similar results can also be found in a serial of previous literatures reporting a high selectivity for $2e^-$ -ORR at 0.7 V equilibrium potentials (*e.g.*, *J. Am. Chem. Soc.* 141, 12372-12381 (2019) and *Adv. Energy Mater.* 10, 2000789 (2020)).

Comment 3: Ordinary N-doped carbon is known not to be selective to the $2e^-$ pathway in alkaline media. [10.1016/j.coelec.2020.08.015] On the contrary, the authors report a very high selectivity to the $2e^-$ pathway in alkaline media. This point should be more carefully clarified. The authors propose multiple pyrrolic nitrogen sites but the existence of such sites are not supported by experimental data. As to the synthetic method, the authors use melamine for the N source, but there is no particular reason for the formation of such N species.

Response: We would like to thank the reviewer for this very insightful comment. Indeed, some previous publication mentioned that nitrogen-doped carbon materials did not show a high selectivity for $2e^-$ -ORR. However, the nitrogen species on these reported materials are almost graphitic nitrogen and pyridinic nitrogen types, which experimentally exhibited good $4e^-$ -ORR properties. This is also in agreement with our previous findings (*e.g.*, *Angew. Chem. Int. Ed.* 51, 3892 (2012) and *J. Am. Chem. Soc.* 133, 20116-20119 (2011)).

However, in other references published recently (*e.g.*, *J. Mater. Chem. A* 10, 4749 (2022) and *Adv. Energy Mater.* 10, 2000789 (2020)), it has been suggested that pyrrolic nitrogen, the five-membered ring configuration nitrogen present only at defects over carbon frameworks, may exhibit certain selectivity towards $2e^-$ -ORR. To more clearly express this, we have further modified the manuscript on Page 5, as follows:

“Compared with others N doping configurations with $4e^-$ -ORR properties (*e.g.*, pyridinic N or graphitic N), which often possess excessively stronger adsorption for $*OOH$, the pyrrolic N configuration with relatively weaker adsorption towards $*OOH$ should be a more suitable candidate

for constructing the proposed multiple-site catalysts.”

For the comment that melamine does not specifically produce pyrrolic nitrogen, we do agree with the reviewer at this point. It needs to be notified, however, the carbon nitride produced by melamine also serves as a sacrificial template for the *in-situ* formation of graphene-like carbon nanosheets, in addition to a precursor for nitrogen doping. The high pyrrolic nitrogen in the material is thus reasonably associated with the material’s defect-rich feature, which provides a feasible environment for pyrrolic nitrogen doping. To more clearly express this point, we have added the corresponding discussion in the manuscript on Page 6, as follows:

“Melamine was subsequently added to the solution. C_3N_4 produced during high-temperature treatment of melamine as the nitrogen precursor and a secondary template to form graphene over the resol-derived mesoporous carbon.”

Due to the limitation in the current material characterization techniques, we are unfortunately unable to directly observe the multiple pyrrolic nitrogen sites. However, such configurations can be indirectly evidenced considering the structural/chemical characterization and theoretical calculation results. Firstly, the sharp rise in the low-pressure region in the nitrogen adsorption-desorption isotherms shows the presence of a large number of micropores in the material (<2 nm). Accordingly, we have constructed the material’s possible single-unit structural models on the basis of the atomic proportions of C (gray balls) and N (blue balls). Even in an extreme scenario where each nanopore (2 nm) is surrounded with only one layer of six-membered rings in each repeating unit (Figure S12a-d, *i.e.*, the most extremely porous case that may exist, with the largest possible amounts of pores on materials), three pyrrolic nitrogen still exist at the edge of the pore. In this case, adjacent multiple pyrrolic nitrogen structures still exist in three of the above four cases. In other cases, where the pore/carbon ration is slightly lower than these extreme cases, multi pyrrolic nitrogen configurations appear almost inevitably, as shown in Figure S12e. Consequently, the proposed multi pyrrolic nitrogen configurations should be almost definitely to exist in our material from a statistic point of view.

Figure S12 a-e) Simulation of the distribution of pyrrolic nitrogen atoms in different substrates.

To more clearly express this point, we have added to the above discussion in the revised manuscript on Page 16, as follows:

“The content of melamine during the material’s fabrication affects the carbon nitride intermediate product, which can significantly alter the structure of obtained carbon material as well as the nitrogen dopant types on it. On this basis, the dual templates utilized in this work would collaboratively generate sufficient pores that provide edge-like defects in the carbon framework of the obtained material, as the favorable environment for pyrrolic N dopants. Combining these structural and chemical characterization results, when the ratio of C atoms to pyrrolic N atoms is 20:1, adjacent multiple pyrrolic N structures should be almost certainly presented in the material, forming the expected clusters containing multiple pyrrolic nitrogen sites for $2e^-$ -ORR (details discussion can be found below Figure S12 in the supplementary information).”

Comment 4: The authors use the DFT study to explain the high catalytic activity, but this part is not convincing. The proposed active sites have not been supported by experimental data, and this method does not consider the effect of pH.

Response: We thank the reviewer for this insightful and constructive comments. In the above response, it is proved that such pyrrolic nitrogen clusters should exist in the material from a static point of view. On this basis, we further calculated the binding energies of three typical dual-pyrrolic

nitrogen doping configurations, and their binding energies are on a similar level and less than 0. This again proves that such multiple pyrrolic nitrogen conformation is thermal dynamically possible to exist in the material. It is worth mentioning that the structures of these models keep stable at 300 K AIMD (Figure S26), indicating their high possibility of actually exist under ambient condition. Meanwhile, the $2e^-$ -ORR overpotentials of all the three models are very low, and the model with the lowest overpotential is chosen for the subsequent calculations.

Figure S26 a-c) Dual-pyrrolic nitrogen active sites with different structures.

To more clearly explain this, according to the valuable suggestions of the reviewer, we have also provided additional explanations in the manuscript on Page 17 of the manuscript.

“We have simulated three possible structures, all of which have very low overpotentials by theoretical calculations. In order to facilitate the subsequent calculations, we chose the first structure with the lowest overpotential as the best model.”

In the process of performing kinetic slow growth calculations, we indeed introduced OH^- to more realistically simulate the process under alkaline conditions, and the specific parameters can be found in the method section of the paper (Page 23).

“The whole system contains 45 H_2O molecules (two layers of water molecules on the surface of N-doped graphene) and one OH^- in the bulk water to model the alkaline environment. In this case, the OH^- concentration is high enough to ensure a pH value higher than 13 and the density of water can be estimated to be $\sim 1 \text{ g cm}^{-3}$.”

Response to Reviewer 2#

Comment 1: In this work, Peng et al. reports the synthesis of a new carbon based nanomaterial for the electrocatalytic O₂ reduction. Apparently the work claims high H₂O₂ formation rates, but I have doubts about the reliability of the experiment. A current density of 40 mA cm⁻² is extremely high, very strange for a purely carbon based catalyst, and it is very odd that such high currents are not reflected in the RRDE tests, where rather typical current density are observed, even though the catalyst is the same. Why such a big discrepancy?

Response: Firstly, we would like to thank the reviewer for the comments and valuable suggestions on this manuscript, which are very helpful in improving the quality of our work.

During the RRDE test, the powdery catalysts were formulated into a slurry, and coated subsequently on the rotating ring disc electrode (RRDE) to test the selectivity and electron transfer number of the material. In this case, the diffusion limiting current, *i.e.*, the plateau in the linear scanning voltammogram, is quickly reached, following the kinetic limiting region. This is the highest achievable current at a given rotating speed of the RRDE and electron transfer number of ORR in an oxygen-saturated electrolyte without directly purging oxygen at the electrode during test. Consequently, RRDE is more often used to accurately determine the electron transfer numbers/selectivity of ORR of a material, rather than to probe the H₂O₂ yield.

As for the subsequent H₂O₂ yield test, the catalyst slurry was coated on the surface of the carbon paper, forming a gas diffusion electrode for the test. In this test, oxygen is directly and continuously purged at the surface of the electrode during the test, forming three-phase interface with sufficient oxygen supply. That is why the current density during the H₂O₂ yield test is significantly higher than the value obtained on RRDE. Similar results can be also found in many other previous works, (*e.g.*, *J. Mater. Chem. A* 10, 4749 (2022), *Adv. Energy Mater.* 10, 2000789 (2022), and *Energy Environ. Sci.* 14, 5444 (2021)) where the current density during the H₂O₂ yield test is much higher than the current density in the RRDE test.

In addition, we indeed have excluded the influence of the effect of H₂O₂ generated on the carbon paper in this work by directly subtracting the H₂O₂ yield of pristine carbon paper. Considering the unclear explanation of this issue in the current manuscript, we have revised the manuscript on Page 13, as follows:

“The H₂O₂ yield of P-NMG-X was then evaluated in an H-type cell. In order to maximize the

electrocatalytic three-phase interface, the reaction was performed using P-NMG-X-coated carbon paper as the working electrode, a Pt plate as the counter electrode, and a Nafion film to separate the cathode/anode chambers.³⁷ The H₂O₂ yield of the catalyst was obtained by subtracting the H₂O₂ yield of the carbon paper for the same time from the total yield after the test.”

Comment 2: Moreover, the electrocatalytic experiments are incomplete: first of all, there is some confusion in the chronoamperometric experiments for the 24 h: in the Experimental part it is reported that the i-t curves are recorded for 30 min experiments, this is a very short time; so, are the CA experiments effectively run for 24 h continuously or it is 30 min experiment repeated many times? This is important to know because it has been reported that the selectivity may change after the initial short time.

Response: We are indeed sorry for this unclear description in the manuscript. In exploring the effect of reaction potential on the yield, we used 30-min tests to increase the testing efficiency. For the CA test/durability test, it was continuously run for 24 h without interruption, rather than many repeated 30 min experiments. During this 24 h test, the current remained stable and the H₂O₂ concentration increase linearly with time, clearly demonstrating the excellent stability of our material for long term. In addition, to further demonstrate the stability of the material for repeated operations, we have also conducted 12 rounds of consecutive 30-min tests (Figure S23). As can be seen from the Figure S23, the P-NMG-10 maintained a stable H₂O₂ yield and FE in the 12 rounds of repeatability tests, demonstrating the excellent catalytic performance and stability of the material.

Figure S23 Yield and FE for different rounds.

In addition, we have also modified the manuscript (Page 15), as follows:

“...A continuous 24h non-stop i-t test was conducted...”

“In addition, several rounds of 30 min repeatability tests were conducted, the material exhibited stable yield and FE in all 12 rounds of testing (Figure S23). Both of the above experiments demonstrated the good stability of P-NMG-10.”

Comment 3: Then, a control experiment is missing to check the contribution of the supporting WE. Finally, the study focuses on one material only, where melamine was the N-doped carbon precursor, but the authors have not checked other substrates to see if this high activity arises uniquely from the melamine precursor. For example, in a 2018 study, Iglesias et al. (*Chem.* 4, 106 (2018)) used carbon nanohorns as the template and several N-containing carbon precursor to see the differences in activity.

Response: We thank the reviewer for this valuable comment, which is helpful for improving the quality of the manuscripts. As mentioned in the response to Comment 1, control experiment for the supporting WE was indeed conducted to rule out any contribution of the carbon paper substrate in the H₂O₂ yield test.

As for preparing nitrogen-doped carbon materials, they are often obtained by annealing a mixture of carbon and nitrogen precursors. In this process, solid nitrogen-containing materials, such as melamine, dicyandiamide, and cyanamide, are often adopted, which are all converted into carbon nitride at about 550-600 °C and then completely decompose at about 750 °C.

In this work, the main role of melamine is to produce C₃N₄ as a sacrificial template to form graphene-like carbon nanosheets and precursor for nitrogen doping. In this work, the main research objective is to modulate the single pyrrolic nitrogen configuration into multiple pyrrolic ones by controlling degree of defects/microporosity of the carbon substrate. In this regard, we believe that different types of solid nitrogen precursors may have insignificant impact on the process. The work mentioned by the reviewer is indeed helpful for in-depth exploration of nitrogen doping, and has been cited in the revised manuscript. The effect of different nitrogen sources will be further explored in our future work. For the more comprehensive and deep introduction, we have also provided more details on the role of melamine in the revised manuscript on Page 6, as follows:

“Melamine was subsequently added to the solution. C₃N₄ produced during high-temperature treatment of melamine as the nitrogen precursor and a second template to form graphene over the

resol-derived mesoporous carbon.”

Comment 4: Other issues arises from the characterization of the material. Melamine is known to evolve tographitic carbon nitride after high T treatment, and in fact the 398 and 401 eV peaks are also typical of carbon nitride, the authors speculate on the presence of pyridinic and graphitic N, this could be in effect some carbon nitride. The distribution of the three signals is also rather typical, I cannot see how the activity is so much controlled here by the structure, especially because the FE is not any higher (actually even smaller) than many other studies. XANES as well assign pyrrole, pyridine and graphitic, but standards are not reported: the XANES of the three molecules should be measured for undoubtedly assign the peaks. Solid state ^{15}N NMR could be of aid to check the N groups.

Response: As mentioned in the response to the above comment, carbon nitride decomposes completely after the high temperature treatment at 750-800 °C, which is much lower than the annealing temperature of this work (900 °C). Consequently, no carbon nitride should exist in the finally obtained materials in this work and it thus not affect the fitting of the different nitrogen configurations in the high resolution N1s spectrum. Similar results have also been obtained in our previous works (*Adv. Mater.* 26, 6074 (2014), *Adv. Mater.* 25, 6226 (2013), *Angew. Chem. Int. Ed.* 51, 11496 (2012)).

We also referred to several representative works in the interpretation of peak positions and peak widths of different configurations in the deconvolution of the high resolution N1s spectrum, and we believe that the XPS data in this work have a high degree of confidence. To reassure the accurate interpretation XANE results, we have also further referred to literatures with detailed identification and discussion about the XANES peak positions (*e.g.*, *ACS Mater. Lett.* 4, 320 (2022); *Adv. Mater.* 2110653 (2022); *Adv. Energy Mater.* 10, 2000789 (2020)), which show similar results to this work. Due to high electrical conductivity of the material, which will cause serious interference to the output of solid-state NMR and result in a lack of credibility in the data, we are unable to conduct solid state NMR on it. However, we performed FTIR tests on the P-NMG-5/10/15 materials. The two characteristic peaks (N–H, C–N) of P-NMG-10 corresponding to pyrrolic nitrogen are much stronger than those of P-NMG-5/15. The data of FTIR are consistent with the chemical composition of the materials analyzed by XPS and XANES, proving that P-NMG-10 has the highest content of pyrrolic nitrogen. Corresponding discussions have been added

in the main text (Page 10):

“The FTIR spectra of the materials show a broad band centered at 3430 cm^{-1} corresponding to the N–H stretching vibrations from pyrrolic nitrogen structure (Figure S31). The peak at $1025\text{--}1200\text{ cm}^{-1}$ is assigned to C–N bonds in the molecule. These two FTIR characteristic peaks of P-NMG-10 material is higher than that of P-NMG-5/15, which proves that P-NMG-10 has a higher content of pyrrolic nitrogen.”

Figure S31 FTIR spectra of the P-NMG-5/10/15.

Comment 5: The conclusion that abundant mesopores are confirmed by TEM is a bit of a speculation, there is only one image with three spots that may be random structural features, it does not make a statistic to draw conclusions. Moreover, the pore distribution shows a curve that goes over a very large distribution of the pore dimension, there is not a peak. The N_2 isotherm also is recorded mistakenly, the loop does not close well, indicating problems in the measurements. The material is not fully characterized, a control experiment should check the presence of metal impurities. I do not understand why different melamine loading lead to similar N content.

Response: We would like to thank the reviewer for this insightful comment and careful reviewing. According to the comment, we have further observed different regions of the material under TEM.

As shown, under low magnification, it can be clearly seen that pores of a few nanometers uniformly disperse over the material, mostly over the porous carbon block sections, which agrees well with the hysteresis in the middle pressure region in the nitrogen adsorption/desorption isotherm. Combined with the results of TEM and N₂ sorption, the material has three parts of pore structure, which are micro-pores with diameters of about 1-2 nm, mesopores of 10-20 nm, and some pores larger than 50 nm formed by the stacking of porous carbon blocks or nanosheets. This is also the reason for the wide pore size distribution for the material, rather than a single peak. To more clearly demonstrate the pore structure of the material, we have put these additional TEM images in the supplementary as Figure S5.

Figure S5 a-f) TEM image of P-NMG-10.

As for the unclosed nitrogen adsorption-desorption isotherms, we also noticed this phenomenon. Repeated nitrogen sorption tests have been conducted and similar results were obtained. This might be attributed to several reasons. Firstly, because of the soft template used in this work, there might be some flexible micro pores existing in the carbon materials. They may shrink during the nitrogen sorption test, leading to incomplete N₂ desorption and the non-closure of isotherms. Similar results have also been found in mesoporous carbons prepared by the similar soft template method (*Nat. Commun.* 4, 2798 (2013), *Angew. Chem. Int. Ed.* 44, 7053 (2005), *Angew. Chem. Int. Ed.* 50, 5947 (2011), *Carbon* 47. 2688 (2009)). Secondly, the multi pyrrolic nitrogen

clusters in the material may also cause chemical adsorption of N₂ gas. In previous researches, carbon nitride with cluster-like nitrogen species presented a high activity for electro/photocatalytic nitrogen fixation, in which the chemical adsorption of N₂ on the multi nitrogen clusters on the material is involved as the first step (*Angew. Chem. Int. Ed.* 57, 10246 (2018), *Nano Lett.* 20, 4, 2879 (2020), and *J. Mater. Chem. A* 8, 13292 (2020)). Regarding this, chemical adsorption of nitrogen may occur on our material as well during the test, leading to incomplete desorption.

Taking into account the reviewers' comment, we will also provide additional explanations in the main text on Page 7, as follows:

“It is worth noting that the adsorption and desorption curves appear to be non-closed, which may be caused by the flexible pore structure or N₂ chemisorption on the material during the test.”

Regarding the reviewer's concern about trace metals, we have conducted ICP-MS test of the material and the results are listed below. As shown, most of the metal elements in the material are negligible, with a weight content below 1 ppm (10⁻⁴ wt.%). As for Fe, Mn, and Ni, their contents are still ≤ 0.01 wt.%, which can also be regarded as negligible for providing any electrochemical activity, considering their values that are even much lower than the typical single atom catalysts (1-5 wt.%). Considering this, it is reasonable to believe the metal impurities in the material, if any, should not affect its electrochemical performance, considering their extremely low amount. Corresponding discussions have been added in the main text (Page 11) and SI (Page 20).

“In addition, ICP-MS characterization of P-NMG-10 was performed in order to exclude the possible interference of trace metals in the material for subsequent electrochemical tests. The results of Table S3 demonstrate that P-NMG-10 does not contain common transition metals and noble metals.”

Table S3 The ICP-MS of P-NMG-10.

Element	Concentration	Unit	Element	Concentration	Unit
Ag	0.0099	mg/kg	Ni	66.5440	mg/kg
Au	0.0106	mg/kg	Pd	0.0541	mg/kg
Bi	0.2592	mg/kg	Pt	0.5546	mg/kg
Co	0.0325	mg/kg	Re	0.2399	mg/kg
Cr	0.2281	mg/kg	Rh	0.0238	mg/kg
Cu	2.6291	mg/kg	Ru	0.1644	mg/kg
Fe	12.6863	mg/kg	Ta	0.1813	mg/kg

Hg	0.4056	mg/kg	Ti	3.8156	mg/kg
Ir	0.1616	mg/kg	V	0.1359	mg/kg
Li	3.9857	mg/kg	W	0.3043	mg/kg
Mn	125.8709	mg/kg	Zn	0.7164	mg/kg
Mo	0.1651	mg/kg			

As for the last comment about the nitrogen content of the material, they are not exactly the same for the serial of materials. Actually, with the increase of melamine content in the precursor mixture, the nitrogen content of in the finally obtained material increases, but relatively slow. This might be attributed to the fact that resol, the carbon precursor, was coated outside melamine particles during the fabrication of the material. The very good contact of these two precursors guarantees the efficient doping of nitrogen in the final carbon material. Even the material obtained from the lowest melamine precursor (*i.e.*, P-NMG-5) has a high nitrogen content, and further increasing the melamine ratio increases the nitrogen content in the final materials, but only slightly.

However, it is still worth mentioning that, apart from simply doping carbon, melamine also forms carbon nitride during the fabrication of material, which acts a sacrificial template for forming graphene-like carbon nanosheets and may affect the overall content of defects in carbon as well. The content of melamine thus determines the overall defect degree in the material, which is essential for the formation of pyrrolic nitrogen in carbon framework and has been reported in previous works as well (*J. Am. Chem. Soc.* 140, 12469 (2018), *Small* 18, 2104941 (2022), *Angew. Chem. Int. Ed.* 57, 16898 (2018)).

Comment 6: In the table of comparison, it is not indicated to which materials the authors are comparing their activity, and some carbon based catalysts which are among the best catalysts are not even reported in the manuscript. Overall, this work leaves me with many doubts, and I think that more experimental work must be conducted to undeniably understand the factors that determine such high currents, surely at this stage publication in Nat comm cannot be recommended.

Response: According to the comments of the reviewer, more details about the materials for comparison and the corresponding references have been listed in table S2 of the SI. To more clearly illustrate the comparison, we have also modified the manuscript on Page 14, as follows:

“To better illustrate the outstanding performance of this material, it is then compared with the recently reported $2e^-$ -ORR catalysts in terms of H_2O_2 yield and FE (Figure 4d and table S2).”

Figure 4 $2e^-$ -ORR catalytic performance of P-NMG-X materials in the two-compartment cell. a H_2O_2 yield of the materials tested at 0 V vs. RHE. b The H_2O_2 yield of P-NMG-10 at different potentials. c The long-range LSV curves of P-NMG-10 in O_2 and Ar-saturated electrolytes. d Comparison of the H_2O_2 yield and FE of P-NMG-10 with some of the reported samples. e The chronoamperometry testing curves and linear fitted lines for the change in H_2O_2 concentration.

Table S2 The electrochemical performance comparison with electrocatalysts recently reported.

Catalysts	Potential (V vs. RHE)	FE	H_2O_2 yield ($mol\ g^{-1}\ h^{-1}$)	Ref.
P-NMG-10	0	90	12	This work
P-NMG-10	-0.6	80	30	This work
G-COF-950	0.1	69.8	1.29	1
N-doped porous carbon	0.1	55	0.46	1
N-doped porous carbon	0.2	58	0.35	1
N-doped mesoporous carbon	0.1	70	0.56	2
Biomass derived N/C	-0.67	51	0.05	3
a-NiO NSs	0.45	85	0.23	4
Co ₁ -NG(O)	0.6	95.6	0.42	5
Oxo-G/ $NH_3 \cdot H_2O$	0.2	43.6	0.22	6
rGO-PEI	0.74	90.7	0.11	7
Ni- N_2O_2	~0.4	90	5.9	8
N-FLG-8	1.8 (cell voltage)	90	9.66	9

Single site Co-NC	0.1	52	0.19	10
N-O-P-C-800	0.25	65	1.47	11

Response to Reviewer 3#

Original comment: In the manuscript the authors introduce N-doped carbon-based material, which was synthesized from resol and melamine at various ratios. The authors claim that the products of synthesis (achieved at 900 °C and under N₂ atmosphere) differed in nitrogen content and chemical composition, which information was derived from HR XPS spectra. The material was characterized by a wide spectrum of experimental characterization techniques and described as graphene/mesoporous carbon composite. The Raman spectra indicate on very defective and functionalized carbon structure. The synthesized materials were used for catalysis of hydrogen peroxide production via 2e⁻-ORR process. The authors claimed they observed record yield for hydrogen peroxide production reaching up to 30 mol/g h and sufficient Faradaic efficiency (in alkalic media). They also provide theoretical calculations to support the claims of the study, that pyrrolic sites are responsible for this activity. Without any doubt the work is interesting addressing hot-topic research focused on hydrogen peroxide production over carbon-based catalysts. On the other hand, there are many critical points, which have to be clarified before publication of this manuscript.

Comment 1: Characterization and elemental composition. The authors provide survey XPS spectra of the materials, but they do not provide elemental composition in the main text (just content on nitrogen content). They should comment particularly on oxygen content, which is quite high and pronounced also in the Raman spectra. They should also clarify, how the HR XPS spectra of N have been fitted (A peak position assignment is not clear. They refer to a work, where the peaks were assigned ad hoc. The authors must provide convincing peak assignment for N-doping in graphene. In addition, they should explain which procedure has been used for spectra deconvolution.). The authors have to estimate an error of the fitting. The role of oxygen and the variability in oxygen content (Table S1) should also be clarified.

Response: Firstly, we would like to thank the reviewer for his/her time and effort in reviewing this manuscript, together with the valuable comments and positive recommendations.

The deconvolution of all XPS spectra in this work was performed using Avantage software. Indeed, we agree with the reviewer that all elemental composition obtained for XPS spectra should be provided in the main text. Therefore, we have added the following discussion (Page 9) and

corresponding table (Figure 2b, Page 9).

“Similar to the EDS results, C, N, and O elements can be found in the survey scan of P-NMG-5,10 and 15 (Figure S9).”

Figure 2 Chemical structure characterization and XANES measurements of P-NMG-X. a TEM image and the corresponding TEM-EDS elemental mapping of P-NMG-10. b Atomic percentages of P-NMG-X analogues. N 1s XPS spectra for c P-NMG-5, d P-NMG-10, and e P-NMG-15. f The relative contents of graphitic nitrogen, pyrrolic nitrogen, and pyridine nitrogen species in the materials. g N K-edge XANES spectra of the materials.

As commented by the reviewer, the XPS and Raman analysis concluded the existence of oxygen in the materials. It can be found that P-NMG-0 possessed a higher oxygen content than the other three, which however show a similar level of oxygen content. The oxygen species should originate from the phenolic resin in the carbon precursor, and the highest content for P-NMG-0 should be attributed to its bulky structure without large pores for the release of oxygen-containing species during high temperature annealing. The relatively lower oxygen content for P-NMG-5/10/15 may be due to their abundant macropores as well as the melamine precursor that releases ammonia and other nitrogen-containing species as possible reduction agents to help remove oxygen from carbon. Corresponding discussions have been added in the main text (Page 9) :

“It can be seen from Figure 2b that the oxygen content in P-NMG-0 is much higher than that in P-NMG-5/10/15. The highest content for P-NMG-0 should be attributed to its bulky structure without large pores for the release of oxygen-containing species during high temperature annealing. The relatively lower oxygen content for P-NMG-5/10/15 may be due to their abundant macropores as well as the melamine precursor that releases ammonia and other nitrogen-containing species as possible reduction agents to help remove oxygen from carbon.”

As for the error in the fitting of the XPS result, there is currently no standard descriptor to evaluate this. However, this can be reasonably assessed by the difference of the envelop of the deconvoluted peaks and the actual XPS curve.

This can be calculated by:

$$\text{Error} = \frac{|\text{Envelope} - \text{Counts}|}{\text{Counts}}$$

Figure S30 N1s peak fitting error of P-NMG-5/10/15.

As shown, the calculated data of N1s XPS spectrum of P-NMG-X is always below 3.4% for this work, suggesting the error in the XPS deconvolution in this work is fairly small. In addition, the N 1s spectrum of the material can be deconvoluted into three peaks at 398.3, 399.8, and 401 eV. This is also in good alignment with the previous reported positions of pyridinic-N (398.5 eV), pyrrolic-N (400.1 eV), graphitic-N (401.1 eV) (*Science* 351, 61 (2016), *Adv. Energy Mater.* 10, 2000789 (2020) and other literatures). Therefore, we believe that the XPS data fitting of this work

should be reasonable.

On the other hand, the oxygen content of all P-NMG-5/10/15 is almost identical and lower than the nitrogen content. There is no apparent dependence between the ratio of oxygen-containing functional groups fitted in Table S1 and their experimentally tested catalytic properties. Considering these factors, we speculate that elemental oxygen may not be involved in the catalytic reaction. Corresponding discussions have been added in the revised manuscript (Page 10):

“By fitting the split peaks of the high-resolution O1s spectra, there is no significant correlation between the oxygen-containing functional groups and the catalytic properties. So we speculate that oxygen-containing species may not be involved in the catalytic reaction.”

Comment 2: The authors have to provide FTIR spectra documenting suggested chemical composition of the materials. The authors must provide ICP-MS data and determine metal residues, which can significantly contribute to the material performance. They have to provide material characterization after its usage. (One additional point, there seems to appear a small peak corresponding to nitrogen in .0 material (Fig. 2b). Could you rule out nitrogen presence in this material?)

Response: We thank the reviewer for this constructive suggestion. Firstly, we performed FTIR tests on the P-NMG-5/10/15 materials. The two characteristic peaks (N–H, C–N) of P-NMG-10 corresponding to pyrrolic nitrogen are much stronger than those of P-NMG-5/15. The data of FTIR are consistent with the chemical composition of the materials analyzed by XPS and XANES, proving that P-NMG-10 has the highest content of pyrrolic nitrogen. Corresponding discussions have been added in the main text (Page 10):

“The FTIR spectra of the materials show a broad band centered at 3430 cm^{-1} corresponding to the N–H stretching vibrations from pyrrolic nitrogen structure (Figure S31). The peak at $1025\text{--}1200\text{ cm}^{-1}$ is assigned to C–N bonds in the molecule. These two FTIR characteristic peaks of P-NMG-10 material is higher than that of P-NMG-5/15, which proves that P-NMG-10 has a higher content of pyrrolic nitrogen.”

Figure S31 FTIR spectra of the P-NMG-5/10/15.

Regarding the reviewer’s concern about trace metals, we have conducted ICP-MS test of the material and the results are listed below. As shown, most of the metal elements in the material are negligible, with a weight content below 1 ppm (10^{-4} wt.%). As for Fe, Mn, and Ni, although their contents are still ≤ 0.01 wt.%, which can also be regarded as negligible for providing any electrochemical activity, considering their values that are even much lower than the typical single atom catalysts (1-5 wt.%). Considering this, it is reasonable to believe the metal impurities in the material, if any, should not affect its electrochemical performance, considering their extremely low amount. Corresponding discussions have been added in the main text (Page 11) and SI (Page 20).

“In addition, ICP-MS characterization of P-NMG-10 was performed in order to exclude the possible interference of trace metals in the material for subsequent electrochemical tests. The results of Table S3 demonstrate that the metal contents in P-NMG-10 are almost negligible.”

Table S3 The ICP-MS of P-NMG-10.

Element	Concentration	Unit	Element	Concentration	Unit
Ag	0.0099	mg/kg	Ni	66.5440	mg/kg
Au	0.0106	mg/kg	Pd	0.0541	mg/kg
Bi	0.2592	mg/kg	Pt	0.5546	mg/kg
Co	0.0325	mg/kg	Re	0.2399	mg/kg

Cr	0.2281	mg/kg	Rh	0.0238	mg/kg
Cu	2.6291	mg/kg	Ru	0.1644	mg/kg
Fe	12.6863	mg/kg	Ta	0.1813	mg/kg
Hg	0.4056	mg/kg	Ti	3.8156	mg/kg
Ir	0.1616	mg/kg	V	0.1359	mg/kg
Li	3.9857	mg/kg	W	0.3043	mg/kg
Mn	125.8709	mg/kg	Zn	0.7164	mg/kg
Mo	0.1651	mg/kg			

We also agree with the reviewer that the materials stability should be further investigated. However, to assemble an electrode, the materials were mixed with Nafion, made into paste, and coated on carbon paper. It is difficult to recover it for direct chemical characterization after the testing and the Nafion binder would significantly interrupt the fine surface chemistry investigation of the used electrode (*e.g.*, XPS). Therefore, we performed several experiments to demonstrate the stability of the material before and after the electrocatalysis. Firstly, we achieved stable current density and linear cumulative product concentration in long cycle tests up to 24 h. In addition, several rounds of 30 min repeatability tests were conducted, and the material exhibited stable yield and FE in all 12 rounds of testing. Both of the above experiments demonstrated the good stability of P-NMG-10. Based on the above considerations, our catalysts showed satisfactory stability.

In order to exclude the nitrogen interference of P-NMG-0 in Figure 2b, we repeated retested XPS, and no signal for the N element was detected.

Figure S9 XPS full spectrum of P-NMG-X.

Comment 3: Mechanistic insights: The authors do not provide any explanation or testable hypothesis concerning the evolution of various nitrogen forms. Why the material ..10 displays the highest content of pyrrolic nitrogen (double with respect of both “next-door” ..5 and ..15 materials)? This important finding is not clarified.

Response: We would like to thank the reviewer for this very insightful comment. Pyrrolic nitrogen, with five-membered ring structure, exists only at the edge of the carbon framework or around the pores inside the carbon framework, *i.e.*, the defect sites on carbon. Therefore, the content of pyrrolic nitrogen species tends to be proportional to the defects degree of the carbon material, such as for the case of P-NMG-10 sample in this work. This is also reported in previous literatures (*J. Am. Chem. Soc.* 140, 12469 (2018), *Small* 18, 2104941 (2022), *Angew. Chem. Int. Ed.* 57, 16898 (2018)). In these works, the materials with the highest pyrrolic nitrogen content all exhibited the highest I_d/I_g values, *i.e.*, the most defective ones.

As for the highest defect degree for P-NMG-10, it should be the compound result of several factors. On the one hand, the material is composed of two types of carbon structures, including the porous carbon blocks derived from phenolic resin and the none-porous graphene-like carbon nanosheet (Figure S5). In this regard, the P-NMG-0 only contains porous carbon blocks, while P-NMG-15 possess the largest content of graphene. On the second hand, the melamine particles of a few microns in size, which embeds in the phenolic resin precursor to form a composite precursor for the subsequent annealing, also acts a template to form large pores in the porous carbon blocks in the finally obtained material. According to our previous work (*Adv. Mater.* 25, 6226 (2013), *Adv. Mater.* 26, 6074 (2014), and *Carbon* 144, 798 (2018)), such macropores are essential for the efficient release of the F127 soft template during annealing, which, otherwise, is hard to be fully released and may block in the micropores of the porous carbon block. Consequently, P-NMG-0, forming large carbon blocks without any macropores, possess only a very low specific surface area as well as a small content of micropores. In contrast, the carbon blocks in P-NMG-15, with the most abundant macropores, should possess the highest amount of micropores in its carbon blocks. However, the large portion of none-porous graphene-like nanosheet may compromise its overall pore amount. Thus, P-NMG-10, with both a high content of porous carbon blocks and a high porosity of porous carbon blocks, possesses the highest overall amount of micropores, *i.e.*, the most sufficient defect sites for the formation of pyrrolic nitrogen species.

Figure S5 a-f) TEM images of P-NMG-10.

To more clearly express this point, we have included the corresponding discussion of it in the text (Page 11), as follows:

“The highest defect degree for P-NMG-10 should be the compound result of several factors. On the one hand, the material is composed of two types of carbon structures, including the porous carbon blocks derived from phenolic resin and the nonporous graphene-like carbon nanosheet (Figure S5). As the amount of melamine increases, the thickness of the porous carbon blocks decreases and the amount of non-porous graphene-like carbon nanosheet increases. On the other hand, the melamine particles of a few microns in size, which embeds in the phenolic resin precursor to form a composite precursor for the subsequent anneal, also acts a template to form large pores in the porous carbon blocks in the finally obtained material. These macropores can not only affect the specific surface area and defect degree of the material, but also promote the efficient removal of the F127 soft template during annealing. Consequently, P-NMG-0, forming large carbon blocks without any macropores, possess only a very low specific surface area as well as a small content of micropores. In contrast, the carbon blocks in P-NMG-15, with the most abundant macropores, should possess the highest amount of micropores in its carbon block sections. However, the large portion of nonporous graphene-like nanosheet may compromise its overall pore amount. Thus, P-NMG-10, with both a high content of porous carbon blocks and a high porosity of porous carbon blocks, possesses the highest overall amount of micropores, *i.e.*, the most sufficient defect sites for the

formation of pyrrolic nitrogen species. Pyrrolic nitrogen, with five-membered ring structure, exists only at the edge of the carbon framework or around the pores inside the carbon framework, i.e., the defect sites on carbon. Therefore, the content of pyrrolic nitrogen species tends to be proportional to the defect degree of the carbon material. Subsequent chemical structure characterization results demonstrated the highest pyrrolic nitrogen content corresponding to the highest defect level of P-NMG-10.”

Comment 4: Performance: The authors state “The above electrochemical tests clearly demonstrate that P-NMG-10 has excellent $2e^-$ -ORR selectivity, associated with its high pyrrolic N contents.” The authors do not provide convincing data for this strong claim of causation. They just provide some data indicating on a correlation, which is not convincing taking into account the questions concerning material characterization stated above. The authors do not provide any data about catalyst reusability, which is absolutely critical to provide convincing evidence that the material behaves like a catalyst not being consumed during “catalysis”.

Response: We thank the reviewer for this valuable suggestion, which is undoubtedly helpful to improve the quality of our work. We do agree with the reviewer that the convincing evidence should be provided in order to more clearly prove the catalytic activity of the materials.

As stated in the response to comment 1, more convincing results about the analysis of XPS results have been provided, which confirms the highest content of pyrrolic nitrogen in P-NMG-10. In addition, the electrochemical analysis in this work thus shows a positive correlation of the material’s performance with its pyrrolic nitrogen content. In addition, previous works also reported that isolated pyrrolic nitrogen dopants may prefer to catalyze ORR via the $2e^-$ pathway (*Adv. Energy Mater.* 10, 2000789 (2020)), while carbon doped with pyridinic nitrogen catalyzes ORR via the $4e^-$ one (*Science* 351, 61 (2016)). However, we do agree with the reviewer that the claim here might be too strong before discussing the DFT calculation results. Consequently, we have modified the manuscript (Page 13) as follow:

“The above electrochemical tests clearly demonstrate that P-NMG-10 has excellent $2e^-$ -ORR selectivity, which is positively correlated with its high pyrrolic N contents.”

DFT (Page 18): “To better understand this positive relation of material’s performance with its high pyrrolic N contents, density functional theory (DFT) calculations were performed to obtain the adsorption free energy of key intermediates in ORR by an implicit solvation method. Meanwhile,

the ab initio molecular dynamics (AIMD) simulations were also conducted in the alkaline environment to describe the reaction kinetics.”

As for the reusability of the material, we have conducted additional tests with 12 consecutive rounds of electrocatalysis. Specifically, the material was tested for 30 min in each round and the H₂O₂ yield and efficiency were assessed. After that, fresh electrolyte was used in the next round of test and this was repeated for 12 times. As shown in Figure S23, the H₂O₂ yield and FE in each round of test are very close, indicating the materials can be repeatedly used for catalyzing 2e⁻-ORR.

Figure S23 Yield and FE for different rounds

To further illustrate this point, we have also modified the manuscript on Page 16, as follows:

“In addition, several rounds of 30 min repeatability tests were conducted, the material exhibited stable yields and FE in all 12 rounds of testing (Figure S23). Both of the above experiments demonstrated the good stability of P-NMG-10.”

Comment 5: Calculations: The authors employ GGA functional (PBE) with an empirical dispersion term (D3). It is well known that GGA functionals significantly underestimate reaction barriers due to SIE. The author must provide more robust data using methodology/functional, which does not suffer from SIE. The authors have to estimate free energy error bars and comment on convergence of the provided AIMD simulations. Could you provide real water density used in the simulations?

Response: We would like to thank the reviewer for this insightful comment and valuable suggestion.

Regarding the energy barrier error caused by the PBE generalization, in this paper, because of the huge difference between the two (0.92 eV for the $4e^-$ energy barrier and 0.54 eV for the $2e^-$ energy barrier), even if the calculation is changed to another generalization, it does not affect the relative conclusion. The convergence criteria for the relevant calculation parameters of AIMD have been added, and the real water density used in the simulation has been adjusted to 1 g cm^{-3} .

Comment 6: Quality of presentation: The authors provide many features on figures, which are not explained or documented (*e.g.*, Fig. 1b and 1d – there are highlighted areas with dashed blue line, which are not explained). Fig. 4d caption can provide link to Table S3 (the authors should provide complete overview of recent works dealing with hydrogen peroxide production over N-doped graphenes). What are the yellowish vials in inset of Fig. 4e?

Response: We thank the reviewer for his/her very careful reviewing on our manuscripts. We are sorry that some features on figures are not fully described in the text. The blue dashed lines in Fig. 1b and 1d are to draw the boundary between the mesoporous carbon and the graphene film. The yellowish vials in the inset of Figure 4e refer to the presence of hydrogen peroxide in the electrolyte after the reaction by the chronoamperometry reaction. The darker the yellow color, the higher the concentration of H_2O_2 in the electrolyte. This color reaction can visually show the accumulation of H_2O_2 in the electrolyte. All of the above descriptions have been added in the revised manuscript (Page 7 and 16), as follows :

“The boundary of the mesoporous carbon and graphene can be clearly observed, as marked by the blue dashed line in Figure 1b.”

“For P-NMG-10, the ORR current is fairly stable during the whole 24 h testing period. The electrolytes after different reaction periods were subjected to chronoamperometry reactions, and the significant increase of hydrogen peroxide concentration with time can be seen in the yellowish vials in inset of Figure 4e. An average H_2O_2 concentration increasing rate of $304.7 \text{ mg L}^{-1} \text{ h}^{-1}$ and accumulative H_2O_2 concentration up to 7.2 g L^{-1} were achieved, which is again the highest reported values by far (Figure 4e).”

Response to Reviewer 4#

Comment 1: This paper reports the multiple pyrrolic N sites decorated on the graphene/mesoporous carbon composite for $2e^-$ -ORR. The authors have done many physical characterizations (TEM, XPS, XAS, *etc.*) and DFT calculations to identify the active sites for $2e^-$ -ORR but the overall electrochemical activities are not very good compared to recent publications. For example, the authors claimed that the catalysts produce ultra-high yield & high FEs for H_2O_2 electrosynthesis. However, this is not a very notable result since the current density only observed $\sim 30 \text{ mA cm}^{-2}$ with FE $\sim 80 \%$. Many of publications ensure $> 50 \text{ mA cm}^{-2}$ operation with $> 90 \%$ yield (*e.g.*, 200 mA cm^{-2} with FE $\sim 90 \%$, *Angew. Chem. Int. Ed.* 61, e202206544 (2022)).

Response: Firstly, we would like to thank the reviewer for his/her constructive suggestions on the manuscript, which are very helpful in improving the quality of our work.

We agree with the reviewer that the current densities are also important performance parameters in the catalytic tests. However, it is unfair to judge the materials' performance solely based on current densities. Firstly, in the literature mentioned by the reviewer (*Angew. Chem. Int. Ed.* 61, e202206544 (2022)), the 1 M KOH was used, which is ten times of the electrolyte concentration of this work. The highly concentrated electrolyte is more favorable for ORR, which is more advantageous in terms of current density. Similar results have been reported in previous works as well (*Nat. Mater.* 10, 780 (2011)). In addition, the areal catalyst loading in the work mentioned by the reviewer is 1 mg cm^{-2} to achieve a current density of 300 mA cm^{-2} , while the catalyst loading used in this work is only 0.02 mg cm^{-2} , which already achieved a current density of 30 mA cm^{-2} . On this basis, the gravimetric activity of our catalyst is significantly better than the work mentioned.

In order to address this point more clearly, we have provided additional information on this catalyst loading in the text (Page 15), as follows:

“A continuous 24 h non-stop i-t test was conducted. For P-NMG-10, a stable current density of 30 mA cm^{-1} was achieved at a catalyst loading of 0.02 mg cm^{-2} during the whole 24 h testing period.”

Comment 2: Also, the following publication reports similar concept related to the H_2O_2 production from pyrrolic-N sites: Li, L. *et al.* Tailoring Selectivity of Electrochemical Hydrogen Peroxide Generation by Tunable Pyrrolic-Nitrogen-Carbon. *Adv. Energy Mater.* 10, 2000789 (2020).

Response: We would like to thank the reviewer for providing us with a very good and valuable work. In this work, the authors have demonstrated through characterization methods, such as synchrotron-based XAS, that the pyrrolic nitrogen configuration acts as an active center and can effectively adsorb intermediate OOH*, thus achieving a good $2e^-$ selectivity.

Different from this literature, our work for the first time reports the synergy of multiple pyrrolic nitrogen species to achieve higher pyrrolic nitrogen content. This synergy can induce much more favorable kinetics for $2e^-$ -ORR than isolated pyrrolic nitrogen or other nitrogen species as previously reported, achieving the highest reported yield of H_2O_2 and the actual cumulative concentration of H_2O_2 so far. Such synergy has also been proven by simulations of the possible multiple pyrrolic nitrogen configuration based on comprehensive experimental characterizations. The thermodynamic and kinetic properties are demonstrated to be superior to those of single pyrrolic nitrogen. Therefore, we believe that our work shows clear innovative to distinguish it from previous publications and has high feasibility for realizing $2e^-$ -ORR for practical applications.

Comment 3: Therefore, this reviewer suggests the authors to do more in-depth studies and submit it to a more specialized journal. Some more specific comments are listed below: If the authors would like to claim the ultrahigh yield, the $2e^-$ -ORR activities must be further studied in 0.5 M KOH or higher concentration to achieve higher current density. What about the electrochemical activities in acidic media, such as 0.5 M H_2SO_4 ? Because H_2O_2 is stable in acidic medium and it's more beneficial to do $2e^-$ -ORR in acidic medium, especially for carbon-based materials.

Response: Thanks for this valuable suggestion. In order to further illustrate the materials' performance, we conducted the tests in 1 M KOH and 0.5 M H_2SO_4 . As shown, a large increase in current density and yield has been achieved in 1 M KOH but a significant decrease in current density and yield occurred in 0.5 M H_2SO_4 . It is consistent with the previously reported results for carbon materials in acidic electrolytes.

Figure S21 a) Current density b) The H₂O₂ yield and FE of P-NMG-10 in 0.1M KOH, 1M KOH and 0.5M H₂SO₄ electrolyte.

Comment 4: The H₂O₂ is not stable in a strong alkaline solution (0.1 M KOH) in nature thus it's hard to accumulate for a long time (even 2 h). Thus, typically the electrolyte needs to be refreshed or requires additives (*i.e.*, H₂O₂ stabilizer) to prevent H₂O₂ decomposition. I could not find any experimental details related to electrolyte refresh or stabilizer. How did authors address these problems?

Response: There is a general problem of carbon-based materials in the 2e⁻-ORR. They commonly achieve high catalytic activity only under neutral and alkaline conditions. Seldom works about metal-free and carbon-based materials have been reported to achieve a good performance under acidic conditions so far.

On the other hand, H₂O₂ is indeed unstable under strong alkaline conditions, but only at higher concentrations (above 5%). In our actual tests, we found that the concentration of hydrogen peroxide in the reacted electrolyte did not significantly decrease drop even after resting for more than a week. In order to more clearly shown this point, we added different concentrations of H₂O₂ to 0.1 M KOH and tested the concentration change after 24 h of resting. It can be seen that, in 0.1 M KOH, an initial H₂O₂ concentration of 7.5 g L⁻¹ (the total accumulated H₂O₂ concentration after 24 h test of our material) can still be maintained at about 91.7% after 24 h, showing the stability of low-concentration H₂O₂ in diluted alkaline electrolyte. In the revised manuscript (Page 15) and SI (Page 16), we have added the discussion of this result and SI:

“Different concentrations of H₂O₂ were kept in 0.1 M KOH and no significant decrease in H₂O₂ concentration was observed even up to 24 h (Figure S29).

Figure S29. The retention rate of different concentrations of H₂O₂ in 0.1M KOH after 24h resting.

Considering the above factors, it is reasonable to deduce that, apart from produce H₂O₂ under acidic conditions, it is also practically and economically feasible to synthesis H₂O₂ via 2e⁻-ORR under alkaline or neutral conditions for on-demand and decentralized applications. For example, fabric bleaching is performed under alkaline conditions. The optical-Fenton reaction in neutral or weak alkaline conditions is also one of the potential applications for the treatment of refractory organics by in situ decomposition of the prepared hydroxyl radicals from H₂O₂. These practical applications can effectively address not only the degradation of carbon-based materials under acidic conditions, but also avoid the problem of H₂O₂ decomposition under strong alkaline conditions.

Lastly, we would like to thank all the reviewers again for their time and efforts in helping us improve the quality of this manuscript. We greatly appreciate this and have carefully modified the article following the comments and questions point by point. We hope the Reviewers could find that our responses are satisfactory and the revised manuscript could meet the standard of *Nature Communications*.

REVIEWER COMMENTS

Reviewer #1 (Remarks to the Author):

I have read other reviewers' comment, the response sheet and revised manuscript carefully. Although the authors have added extra explanation into the revised manuscript, revision in terms of experimental evidence is quite limited. Therefore, this manuscript is substantially same as the original one, and I have no reason to change my recommendation.

Reviewer #2 (Remarks to the Author):

The authors have improved their manuscript following the reviewers comments and clarifying the indicating aspects. Now I can recommend its acceptance.

Reviewer #3 (Remarks to the Author):

The authors well addressed all points and improved the manuscript significantly. I would recommend its publication.

Reviewer #4 (Remarks to the Author):

N/A

Comment 1: Please report mass loadings of the catalyst in the Methods section and report mass yields (in $\mu\text{mol}/\text{h}$). Accordingly, modify the benchmarking to compare mass yields to other catalysts and report similar conditions as necessary (e.g. light intensity, catalyst loading, time).

Response: Firstly, we would like to thank the reviewer/editor for the valuable comments and suggestions on this manuscript, which are very helpful in improving the quality of our work. We understand the reviewer's and editor's concerns about the presentation of the yields in the manuscript, so we have included additional experimental results and corresponding discussions to make it better illustrated.

Firstly, the mass loadings of the catalyst, the area of the carbon paper electrode, and the mass yields of H_2O_2 have been reported in the main text (Page 23, 15).

“...5.0 μL of the catalyst ink (4 mg mL^{-1}) was drop-casted onto the surface of the working electrode (0.5 \times 1 cm^2 of hydrophobic carbon paper)...”

“...In this case, the highest value is achieved at -0.6 V vs. RHE, with an H_2O_2 yield rate up to ca. 30 mol g^{-1} h^{-1} (611.4 $\mu\text{mol h}^{-1}$ with a FE of 80%), significantly superior to the comparative analogues over the whole testing potential range (Figure 4b)...”

For most of the literatures, the detailed testing conditions (*i.e.*, electrode area used for testing) were not reported (**Table R1**), so it is unable to compare the catalytic performance of the materials solely considering the H_2O_2 mass yield. This is also the reason that the unit “mol $\text{g}_{\text{cat}}^{-1}$ h^{-1} ” is frequently adopted in almost all publications to demonstrate the intrinsic activity of the materials.

Despite this, we still tried to collect the yields (mol g^{-1} h^{-1}) from some published works and converted them according to their mentioned loadings to obtain H_2O_2 mass yields ($\mu\text{mol h}^{-1}$). It can be seen from **Table R1** that even though our areal loading and electrode area are lower than the data reported by other works, the H_2O_2 mass yields are still at the highest level. It should be noted that the only report (Ref. 7) with a higher value (1102.5 $\mu\text{mol h}^{-1}$) than this work (611.4 $\mu\text{mol h}^{-1}$) is obtained with a four-time larger electrode and eight times more catalysts.

Table R1. The H_2O_2 yield comparison with electrocatalysts recently reported.

Catalysts	H ₂ O ₂			Mass yield ($\mu\text{mol h}^{-1}$)	Electrode area (cm^{-2})	Ref.
	Potential (V vs. RHE)	yield (mol g^{-1} h^{-1})	Loading (mg cm^{-2})			
P-NMG-10	0	12	0.04	240.5	0.5	This work
P-NMG-10	-0.6	30	0.04	611.4	0.5	This work
G-COF-950	0.1	1.29	0.10	193.5	1.5	1
a-NiO NSs	0.45	0.23	0.25	57.6	1	2
Biomass derived N/C	-0.67	0.05	0.57	N/A	N/A	3
Co ₁ -NG(O)	0.6	0.42	1	420.8	1	4
Ni-N ₂ O ₂	-0.4	5.90	0.20	N/A	N/A	5
Co-N-C	1.5 (cell voltage)	4.33	0.10	N/A	N/A	6
N-O-P-C-800	0.25	1.47	0.19	1102.5	4	7
Co SACs	-0.4	4.50	0.20	N/A	N/A	8
N-FBMC-500	-1.2	3.18	0.20	N/A	N/A	9

Figure S32. Mass yields for different catalyst loadings per unit electrode area.

Considering the editor's and reviewer's concerns about this point, we have conducted the H₂O₂ yield test with different areal catalyst loadings at 0 V vs. RHE. As shown in **Figure S32**, with the increase of areal catalyst loading, the H₂O₂ mass yield increases, but the Faradaic efficiency decreases. This may be attributed to the thickened catalyst coating on the carbon paper electrode, and the catalyst beneath the surface layer cannot effectively contact oxygen, leading to the unfavorable H₂O₂RR that consumes the produced H₂O₂. Considering these results, we believe that an areal loading of 0.04 mg cm⁻² should be the optimal loading for the preparation of H₂O₂ using this material.

In addition, we also notice that, in some of the recent reports, flow cells were adopted for long-term and non-stop test in 0.1 M KOH and at 0 V vs. RHE. To further demonstrate the excellent performance of the material, we also supplemented the H₂O₂ yield test in a flow cell.

Figure S34. 24 h non-stop cumulative H₂O₂ concentration test in a flow-cell.

As can be seen in **Figure S34**, benefitting from the three-phase interface with a more effective oxygen supply, the 24 h non-stop i-t test results of flow-cell are slightly better than our previous test results obtained in H-cell. While maintaining a stable growth rate (383.3 mg L⁻¹ h⁻¹), the H₂O₂ concentration accumulated over 24 h finally reached 9 g L⁻¹, which is higher than that of 7.2 g L⁻¹ of H-cell. This result also demonstrates the excellent performance of P-NMG-10 in electrocatalytically producing H₂O₂ by 2e⁻-ORR. The above results and discussion we will also add to the main text (Page 16) and SI.

*“...In addition, 24 h non-stop i-t tests in the flow-cell achieved a final cumulative H₂O₂ concentration of 9 g L⁻¹ (**Figure S34**), with an average H₂O₂ concentration increasing rate of 383.3 mg L⁻¹ h⁻¹, slightly higher than that of the H-cell. These test results together demonstrate the excellent performance of P-NMG-10 in fabricating H₂O₂ by 2e⁻-ORR...”*

Considering the reviewer's concern about the presentation of the yield test results, we have also modified some of the manuscript (Page 2, 15 and 16).

“...Consequently, an ultra-high H₂O₂ yield of up to 30 mol g⁻¹ h⁻¹ has been achieved on this material...”

“...The H₂O₂ yield of P-NMG-10 is one of the highest among the materials reported so far...”

“...An average H₂O₂ concentration increasing rate of 304.7 mg L⁻¹ h⁻¹ and accumulative H₂O₂ concentration up to 7.2 g L⁻¹ were achieved in the H-cell, which is one of the highest reported values by far (Figure 4e)...”

Comment 2: *Though we understand the experimental difficulties in quantifying the pyrrolic N-sites in the catalysts, we would appreciate some attempts at providing evidence for or justification of the claim of pyrrolic N-sites with XPS or other techniques. This is especially true for Reviewer #1's current report - which notes plainly that experimental support for their participation in the reaction is quite limited. This is an important point for the study as well, given that these sites are referenced in the title.*

Alternatively, if this is not possible, please rephrase the mechanistic aspects of the work to deemphasize the role of pyrrolic nitrogen sites and note that their existence and participation in the reaction mechanism is speculative at this time, especially because oxygen content does not influence the metrics.

Response: Thanks for this valuable suggestion. Accordingly, we have summarized all the characterization results available so far and discussed the possible multiple pyrrolic nitrogen structures.

Firstly, we confirm the existence of pyrrolic nitrogen species using various chemical characterizations, including XPS, FTIR, and XANES (**Figure R1**). Specifically, the XPS results show that P-NMG-10 has the highest pyrrolic nitrogen content with an atomic percentage of 5%, much higher than the other comparative samples (**Figure R1a-f**). We also added more details to the deconvolution of high-resolution XPS results, for example, by adding the error information about the deconvoluted results (all less than 4%, **Figure R1g**). In addition, the FTIR results also clearly show the existence of N-H and C-N functional groups, which typically belong to the pyrrolic nitrogen configurations, and these typical peaks are obviously stronger for P-NMG-10 than the other comparative samples, showing the higher pyrrolic nitrogen content (**Figure R1h**). Moreover, the XANES results also demonstrate that the

pyrrolic nitrogen content of P-NMG-10 is the highest among samples (**Figure 1i**). Therefore, all these results clearly confirm the abundant pyrrolic nitrogen species in P-NMG-10, which can be up to 5 at.%.

Figure R1. **a** XPS survey spectra of P-NMG-X. **b** Atomic percentages of P-NMG-X analogues. **c** N 1s XPS spectra for P-NMG-5, **d** P-NMG-10, and **e** P-NMG-15. **f** The relative contents of graphitic nitrogen, pyrrolic nitrogen, and pyridine nitrogen species in the materials. **g** N1s peak fitting error of P-NMG-5/10/15. **h** FTIR spectra of the P-NMG-5/10/15. **i** N K-edge XANES spectra of the materials.

Due to the limitation in the current material characterization techniques, however, it is extremely difficult or even impossible for us to directly observe the multiple pyrrolic nitrogen sites on the materials. However, such configurations can be indirectly evidenced on the basis of the structural/chemical characterizations and theoretical calculation results. These results are summarized in **Figure R2**.

In the nitrogen adsorption-desorption test (**Figure R2a**), a sharp rise exists in the isotherms in the low-pressure region, which corresponds to a large number of micropores in the materials (<2 nm). This is also in agreement with the pore size

distribution results. Accordingly, we have constructed the material's possible single-unit structural models on the basis of the atomic proportions of C (gray balls) and N (blue balls), with the existence of such small-sized micropores. Even in an extreme scenario where each micropore (2 nm) is surrounded with only one layer of six-membered rings in each repeating unit (**Figure R2b**, *i.e.*, the most extremely porous case that may exist, with the largest possible amounts of pores on materials), three pyrrolic nitrogen atoms still exist at the edge of the pore. In this case, adjacent multiple pyrrolic nitrogen structures would form in three of the above four cases. In other cases, where the pore/carbon ratio is lower than in these extreme cases, multiple pyrrolic nitrogen configurations appear almost inevitably, as shown in **Figure R2c**. Consequently, the proposed multiple pyrrolic nitrogen configurations should almost definitely exist in our material from a statistical point of view.

Figure R2. **a** Nitrogen adsorption-desorption isotherm, and the corresponding pore size distribution of P-NMG-10. **b-c** Possible distribution of pyrrolic nitrogen atoms in different substrates.

Despite these analyses, we do understand the editor's and reviewer's concern about the lack of direct evidence for the multiple pyrrolic nitrogen sites. Consequently, we have revised the presentation in the title and main text to add our statistical speculation based on the above characterization results to the main text and to deemphasize the role of pyrrolic nitrogen sites to a certain extent (Page 1, 17, and 18), indicating that the

multiple pyrrolic nitrogen site is the statistically possible active site.

For the title, it has been revised: *“Kinetically Facilitating Two-electron Oxygen Reduction by Pyrrolic-rich Nitrogen Sites for Ultra-high-yield Electrochemical Hydrogen Peroxide Production”*

“...Considering the results of the carbon substrate structure and chemical characterization, when the ratio of C atoms to pyrrolic N atoms reaches 20:1, adjacent multiple pyrrolic N structures should be highly likely to present on the carbon substrate, forming the expected clusters containing multiple pyrrolic N sites for $2e^-$ -ORR (Figure S12). In this case, it may also exhibit a similar synergistic effect toward ORR, as widely observed in the other metal-based multiple-site systems...”

“...Graphene with dual-pyrrolic nitrogen doping configuration (Dual-PyrN-Gr) was constructed as a simplified model to simulate multiple pyrrolic N sites that may be presented in P-NMG-10...”

Comment 3: *Please modify or perform more detailed DFT simulations. These simulations should be more reflective of the composition of the material which clearly contains oxygen functional groups (Figure S11) and N-sites (Figure 2), and include pH effects in accordance with Reviewer #1's previous requests.*

Response: We appreciate this insightful comment, and accordingly, we have conducted calculations with the consideration of oxygen species on the material.

Figure S33. a-c Dual-PyrN-Gr active sites with different oxygen functional groups on the same side of the nitrogen atom. d-f Dual-PyrN-Gr active sites with different oxygen functional groups on the opposite side of the nitrogen atom.

In the experimental section, it has been demonstrated by XPS that the material contains three types of oxygen functional groups, including C=O, COOH, and COH. We first introduced three oxygen-containing functional groups in the models for DFT calculation. The same functional groups are constructed either on the same or the opposite side of the nitrogen atoms in two different models for their calculations (**Figure S33**).

The calculated formation energy of these configurations and the overpotential for the $2e^-$ -ORR on them are summarized in **Table S4**. As shown, for the C=O and COOH groups, the formation energies of the configurations with them on the same side of the nitrogen atoms (**Figure S33d, f**) are much higher than those with these species on the other side of the nitrogen atoms (**Figure S33a, c**). As for the COH species, the formation energies for the configurations with COH on the same or opposite side of the nitrogen atoms are almost identical (**Figure S33b, e**). These results suggest that the four configurations, as illustrated in **Figure S33a, b, c, and e**, are more likely to be formed from a thermodynamic point of view.

For these configurations, the existence of COH, C=O, and COOH does not affect the overpotential for $2e^-$ -ORR in comparison with the results obtained on the models

without these oxygen functional groups (**Table S4**). Consequently, it can be concluded that the oxygen-containing functional groups should not significantly affect the performance of the materials in this work. This is also in agreement with the experimental results that the comparative analogues in this work have similar content of oxygen functional groups but distinctly different electrochemical performances.

To better demonstrate this point, we have included the above results and discussion in the revised manuscript (Page 18) and SI.

Table S4. Formation energy and overpotential of active sites with different oxygen functional groups.

Active sites with different oxygen functional groups	Formation energy (eV)	Overpotential (V)
C=O (Same side as N atom)	-520.17	0.16
C=O (Opposite side as N atom)	-520.74	0.06
COH (Same side as N atom)	-524.21	0.01
COH (Opposite side as N atom)	-524.24	0.03
COOH (Same side as N atom)	-539.70	0.46
COOH (Opposite side as N atom)	-540.05	0.05

“...In order to exclude the possible influence of oxygen functional groups in the calculation, we introduced three different oxygen functional groups for the calculation based on the Dual-PyrN-Gr doping configuration (**Figure S33**). The thermodynamic calculation results demonstrate that the $2e^-$ -ORR overpotential of the dual-pyrrolic nitrogen doping configuration site is not affected by the oxygen functional groups (**Table S4**)...”

As for the editor's and reviewer's concerns about the possible effects of pH values, we have added the correction of pH for thermodynamic calculations in the revised manuscript. The equilibrium voltages corresponding to the materials are different under

acidic and basic environments, but the overpotential of the respective reactions is not affected. We show this with the free energy profiles for both $2e^-$ - and $4e^-$ -ORR pathways at different pH values (**Figure R3**).

Figure R3. a-b The free energy profiles for both $2e^-$ - and $4e^-$ -ORR pathways at different pH values.

Correspondingly, we modified **Figure 5a** by additionally labelling the pH value environment simulated in the computational process (Page 17).

Figure 5. Theoretical results on the $2e^-$ -ORR activity on different configurations. **a** Free energy profile of $2e^-$ - and $4e^-$ -ORR pathways at the equilibrium potentials of 0.7 V vs. RHE on different N-doped graphene models. **b** Schematic diagram of $*OOH$

adsorption on Dual-PyrN-Gr in neutral and alkaline solutions under equilibrium potentials. Free energy changes as a function of CV along. **c** C-O bond and **d** O-O bond cleavage for the $2e^-$ and $4e^-$ -ORR pathways, respectively. The initial state, transition state, and final state structures are denoted as IS, TS, and FS, respectively. Blue, gray, red and white spheres represent the N, C, O and H atoms, respectively.

In addition, we also considered the effect of pH value on the kinetic calculations, and we introduced OH^- anions during the calculations to simulate an alkaline reaction environment. The formulation of this part is added to the method section (Page 24):

“...The whole system contains 45 H_2O molecules (two layers of water molecules on the surface of N-doped graphene) and one OH^- anion in the bulk water to simulate the alkaline environment. In this case, the OH^- concentration is high enough to be close to the experimental pH conditions...”

We hope that the above additional calculations, related explanations, and changes to the manuscript's presentation can answer the questions and concerns of the reviewer and editor regarding the theoretical calculations.

Comment 4: *We note that recent work has reported similar electrochemical potentials for $2e^-$ oxygen reduction (<https://www.nature.com/articles/s41467-019-11992-2>), especially regarding the $E_{1/2} = 0.7 \text{ V vs RHE}$. Please cite this study and discuss it in the context of the currently proposed mechanism. Additionally, please offer some justification as to how this observation fits into their mechanistic interpretation.*

Response: Indeed, we do agree with the editor that the onset potential of $2e^-$ -ORR is an interesting research point. We have noticed similar results in several recent publications, where the onset potential of the material obtained in the RRDE test is very close to or even slightly higher than the theoretical potential of 0.7 vs. RHE for $2e^-$ -ORR. This is generally considered to represent a highly facile ORR kinetics with negligible overpotential for O_2 -to- H_2O_2 conversion. We have added the corresponding discussion in the revised manuscript, and the related literature (*e.g.*, *Nat. Commun.* 10, 3997 (2019) and *Nat. Catal.* 1, 156-162 (2018)) has been referred to as well (Page 12):

*“...Meanwhile, for P-NMG-10, a disk current density of 1 mA cm^{-2} was achieved at 0.64 V vs. RHE, which is close to the equilibrium potential for $2e^-$ -ORR (*i.e.*, 0.7 V*

vs. RHE), representing a facile ORR kinetics with negligible overpotential for O₂-to-H₂O₂ conversion.^{37,38} ...”

“37. Jiang, K., *et al.* Highly Selective Oxygen Reduction to Hydrogen Peroxide on Transition Metal Single Atom Coordination. *Nat. Commun.* **10**, 3997 (2019).

38. Lu, Z., *et al.* High-efficiency Oxygen Reduction to Hydrogen Peroxide Catalysed by Oxidized Carbon Materials. *Nat. Catal.* **1**, 156-162 (2018).”

The study on the onset potential of 2e⁻-ORR is indeed very important. We will also explore this point in our subsequent works, and this might give us new insights on achieving kinetic enhancement through materials design to further reduce the 2e⁻-ORR onset potential for metal-free carbon-based materials.

Lastly, we would like to thank all the reviewers and the editor again for their time and efforts in helping us improve the quality of this manuscript. We greatly appreciate this and have carefully modified the article following the comments and questions point by point. We hope the reviewers and editor could find that our responses are satisfactory and the revised manuscript could meet the standard of *Nature Communications*.

References:

1. Zhang, J., *et al.* Graphitic N in Nitrogen-Doped Carbon Promotes Hydrogen Peroxide Synthesis from Electrocatalytic Oxygen Reduction. *Carbon* **163**, 154-161 (2020).
2. Li, R., *et al.* Short-range Order in Amorphous Nickel Oxide Nanosheets Enables Selective and Efficient Electrochemical Hydrogen Peroxide Production. *Cell Rep. Phys. Sci* **3**, 100788 (2022).
3. Yang, Y., *et al.* A Biomass Derived N/C-Catalyst for the Electrochemical Production of Hydrogen Peroxide. *Chem. Commun.* **53**, 9994-9997 (2017).
4. Jung, E., *et al.* Atomic-Level Tuning of Co-N-C Catalyst for High-Performance Electrochemical H₂O₂ Production. *Nat. Mater.* **19**, 436-442 (2020).
5. Wang, Y., *et al.* High-Efficiency Oxygen Reduction to Hydrogen Peroxide Catalyzed by Nickel Single-Atom Catalysts with Tetradentate N₂O₂ Coordination in A Three-phase Flow Cell. *Angew. Chem. Int. Ed.* **59**, 13057-13062 (2020).
6. Sun, Y., *et al.* Activity-Selectivity Trends in The Electrochemical Production of Hydrogen Peroxide over Single-Site Metal-Nitrogen-Carbon Catalysts. *J. Am. Chem. Soc.* **141**, 12372-12381 (2019).
7. Zhang, H., *et al.* Electrocatalyst Derived from Fungal Hyphae and its Excellent Activity for Electrochemical Production of Hydrogen Peroxide. *Electrochim. Acta* **308**, 74-82 (2019).
8. Zhao, Q., *et al.* Approaching a High-Rate and Sustainable Production of Hydrogen Peroxide: Oxygen Reduction on Co-N-C Single-Atom Electrocatalysts in Simulated Seawater. *Energ. Environ. Sci.* **14**, 5444-5456 (2021).

9. Bao, Z., *et al.* Synergistic Effect of Doped Nitrogen and Oxygen-Containing Functional Groups on Electrochemical Synthesis of Hydrogen Peroxide. *J. Mater. Chem. A* **10**, 4749-4757 (202).

REVIEWER COMMENTS

Reviewer #1 (Remarks to the Author):

The manuscript has been indeed improved. I believe that this version is acceptable.